# Correlating instruction-tuning (in multimodal models) with vision-language processing (in the brain)

**Subba Reddy Oota**[1*]**, Akshett Jindal**[2*]**, Ishani Mondal**[3]**, Khushbu Pahwa**[4]
**Satya Sai Srinath Namburi**[5]**, Manish Shrivastava**[2]**, Maneesh Singh**[6]
**Bapi S. Raju**[2]**, Manish Gupta**[7]
[1]Technische Universität Berlin, Germany, [2]IIIT Hyderabad, India, [3]Univ of Maryland, USA
[4]Rice Univ, USA, [5]Univ of Wisconsin - Madison, USA, [6]Spector Inc, USA
[7]Microsoft, India
subba.reddy.oota@tu-berlin.de, gmanish@microsoft.com

## ABSTRACT

Transformer-based language models, though not explicitly trained to mimic brain recordings, have demonstrated surprising alignment with brain activity. Progress in these models—through increased size, instruction-tuning, and multimodality—has led to better representational alignment with neural data. Recently, a new class of instruction-tuned multimodal LLMs (MLLMs) have emerged, showing remarkable zero-shot capabilities in open-ended multimodal vision tasks. However, it is unknown whether MLLMs, when prompted with natural instructions, lead to better brain alignment and effectively capture instruction-specific representations. To address this, we first investigate brain alignment, i.e., measuring the degree of predictivity of neural visual activity using text output response embeddings from MLLMs as participants engage in watching natural scenes. Experiments with 10 different instructions (like image captioning, visual question answering, etc.) show that MLLMs exhibit significantly better brain alignment than vision-only models and perform comparably to non-instruction-tuned multimodal models like CLIP. We also find that while these MLLMs are effective at generating high-quality responses suitable to the task-specific instructions, not all instructions are relevant for brain alignment. Further, by varying instructions, we make the MLLMs encode instruction-specific visual concepts related to the input image. This analysis shows that MLLMs effectively capture count-related and recognition-related concepts, demonstrating strong alignment with brain activity. Notably, the majority of the explained variance of the brain encoding models is shared between MLLM embeddings of image captioning and other instructions. These results suggest that enhancing MLLMs' ability to capture task-specific information could lead to better differentiation between various types of instructions, and thereby improving their precision in predicting brain responses. We make the code publicly available[1].

## 1 INTRODUCTION

Brain encoding aims at constructing neural brain activity recordings given an input stimulus. Prior studies in brain encoding have demonstrated that representations from multimodal models, which align multiple modalities (e.g., vision and language), achieve a higher degree of brain alignment for both image-based and text-based representations compared to vision-only models, particularly when using naturalistic image stimuli (Doerig et al., 2022; Wang et al., 2022; Oota et al., 2022b; Popham et al., 2021; Oota et al., 2024b). Specifically, these studies have shown that representations from multimodal models are better at predicting neural responses in the high-level visual cortex as compared to previous vision-only models like convolutional neural networks (CNNs) (Wang et al.,

---

[1]https://github.com/subbareddy248/mllm_instruction_brain

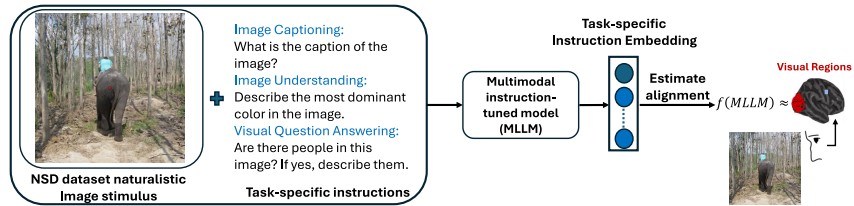

Figure 1: Leveraging instruction-tuned multimodal LLMs for brain encoding with a diverse set of instructions. For the given image, we could obtain different multimodal representations using instructions that ask the model to (i) generate the caption of the image, (ii) identify whether people are present, or (iii) determine the primary colors dominant in the image. Using instruction-specific representations, we estimate the alignment using a simple linear function $f$ (ridge regression) which map MLLM representations to brain recordings.

2022; Popham et al., 2021). However, prior research investigating the effectiveness of multimodal models for brain alignment has primarily relied on image-caption pairs, leaving the full potential of multimodal models, especially when enhanced with large language models (LLM) and task-specific instructions, underexplored. In this paper, we explore how effectively multimodal representations, obtained by prompting MLLMs with various natural instructions, align with visual processing brain regions.

Several previous *unimodal* studies have found that Transformer models finetuned for specific tasks more closely align with brain processes during language comprehension (Oota et al., 2022a; Aw & Toneva, 2023; Sun & Moens, 2023; Oota et al., 2024c), speech processing (Oota et al., 2023; Tuckute et al., 2023; Oota et al., 2024a), and visual processing (Wang et al., 2019; Conwell et al., 2022), yielding better brain predictivity than pretrained models. However, these studies rely on separate task-specific models, which limits generalization, as each task-specific representations are aligned with the same human brain recordings. Furthermore, finetuning data needs to be obtained for each task, and a new model needs to be trained separately. Recently, instruction-tuning has become a widely adopted method for fine-tuning the same baseline large language model across multiple different natural language processing (NLP) tasks. This approach has been shown to outperform task-specific models (Taori et al., 2023; Touvron et al., 2023; Jiang et al., 2023; Abdin et al., 2024; Dubey et al., 2024), and demonstrates improved brain alignment compared to smaller language models (Sun et al., 2023; Sun & Moens, 2023; Loong Aw et al., 2024).

Building on these advances, researchers have extended instruction-tuning to multimodal LLMs (MLLMs) (Xu et al., 2023; Dai et al., 2023; Liu et al., 2024), enabling impressive unimodal and multimodal capabilities. These progressive improvements motivate us to explore the effectiveness of instruction-tuned MLLMs for brain encoding. By leveraging task-specific representations from a single MLLM, we aim to capture multiple aspects of an image beyond simple captioning, such as people, foreground and background elements, interactions between objects, environments, colors, food items, animals, and outdoor scenes. This leads to a critical question related to understanding of human-alignment of AI: *Do these multimodal instruction-tuned models prompted using natural language instructions lead to better brain alignment and differentiate instruction-specific representations?* To address this, we investigate different ways of prompting MLLMs with various task-specific instructions. Overall, this research utilizes various task-specific MLLM representations to develop encoding models based on fMRI responses within a multimodal model framework (see Fig. 1 for workflow).

Using brain recordings of participants watching natural scenes images from the NSD dataset (Allen et al., 2022), we investigate several research questions. First, we explore the effectiveness of MLLM representations and compare their brain alignment with unimodal and multimodal models. For the purposes of this work, we focus on three MLLMs (InstructBLIP (Dai et al., 2023), mPLUG-Owl (Ye et al., 2023) and IDEFICS (Laurençon et al., 2023)), one image-based (ViT-H) and one multimodal model (CLIP). We probe these MLLMs using ten different instructions across six visual tasks. Specifically, we investigate which of these task-specific instructions result in better brain alignment. Second, do the instruction-specific representations from MLLMs differentiate the visual regions that process this information, thereby aligning with the mechanisms of human visual cognition? Third, do task-specific instructions from MLLMs account for visual concepts understanding,

and which brain regions are responsible for processing different visual concepts? Fourth, we use a variance partitioning approach to assess the unique and shared variance of each task-specific instruction to brain responses. This analysis provides insights into how different visual tasks complement or overlap in explaining brain activity, thereby enhancing our understanding of the functional organization of visual processing in the brain.

Our analysis of instruction-tuned MLLMs and brain alignment reveals several key conclusions: (i) MLLMs demonstrate significantly better brain alignment than vision-only models and perform comparably to multimodal models such as CLIP. (ii) While all three MLLMs generate task-specific output tokens based on instructions, not all instructions contribute to higher brain alignment. Specifically, the image captioning instruction leads to stronger brain alignment in regions like the EBA (extrastriate body area), PPA (parahippocampal place area), and FFA (fusiform face area), whereas instructions related to image understanding result in higher alignment in early visual regions. (iii) Furthermore, while MLLMs capture several visual concepts, such as counts and recognition, for other concepts like color, positional understanding, and general scene understanding, MLLMs exhibit similar brain alignment patterns. (iv) By employing a variance partitioning approach, we find that most of the variance is shared across instructions, with a high overlap between Image Captioning (IC) and other prompts but lower overlap with image understanding and scene recognition. This suggests that MLLMs could improve in differentiating between various types of instructions.

Overall, our work is the first to propose the use of instruction-tuned MLLMs and to demonstrate the differences in task-specific representations within MLLMs, along with the reasons behind these differences in relation to brain alignment. We make the code publicly available[1].

## 2 RELATED WORK

**Brain encoding using multimodal models.** The human brain perceives the environment using information from multiple modalities (Gauthier et al., 2003). Therefore, examining the alignment between language and visual representations in the brain by training encoding models on fMRI responses, while extracting joint representations from multimodal models, can offer insights into how our brain processes multimodal information. For instance, it has been shown in Doerig et al. (2022); Wang et al. (2022); Oota et al. (2022b); Popham et al. (2021) that multimodal models like CLIP (Radford et al., 2021b) better predict neural responses in the high-level visual cortex as compared to previous vision-only models. Additionally, Tang et al. (2024) demonstrate the use of multimodal models in a cross-modal experiment to assess how well the language encoding models can predict movie-fMRI responses and how well the vision encoding models can predict narrative story-fMRI. Nakagi et al. (2024) analyzed fMRI related to video content viewing and found distinct brain regions associated with different semantic levels, highlighting the significance of modeling various levels of semantic content simultaneously. However, these studies have primarily focused on multimodal models aligned in embedding space when text captions were provided as input, leaving the new class of instruction-tuned MLLMs-which utilize task-specific natural language instructions—still unexplored. Unlike previous work, we are the first to study multimodal instruction-tuned models with language-guided instructions and to perform comprehensive brain alignment analysis while subjects are engaged in viewing passive images.

**Task-based brain alignment.** Our work is also closely related to that of Wang et al. (2019); Oota et al. (2022a); Aw & Toneva (2023); Sun et al. (2023) and Aw et al. (2023), who propose using task-specific model representations to study the contribution of individual tasks to brain alignment. Wang et al. (2019) investigated 21 computer vision tasks to explore which vision tasks are more aligned with the brain while subjects engaged in viewing passive images. Similarly, Oota et al. (2022a) and Sun et al. (2023) explored 10 GLUE NLP tasks to study which NLP tasks are more brain-aligned during reading and listening to stories. Aw & Toneva (2023) further extended the comparison by evaluating pretrained models that were trained either on web data or BookSum stories to determine whether BookSum models provide better character-specific information in brain alignment tasks. More recent work by Aw et al. (2023) uses instruction-tuned language models to investigate the effect of natural language instruction model representations on brain alignment across layers for language comprehension. We complement these works by examining the impact of a wide range of multimodal instruction-tuned models on brain alignment and by studying the effect of task-specific, language-guided instructions from MLLMs on their alignment with brain activity.

## 3 DATASET AND CURATION

**Brain dataset.** We use the Natural Scenes Dataset (NSD) introduced by Allen et al. (2022), which contains high-quality fMRI readings of 8 participants watching images from the COCO dataset (Lin et al., 2014). We analyzed brain data for 4 participants (who completed all the sessions) where each participant was presented with $10,000$ images in total ($1,000$ common for all participants and $9,000$ unique for a participant). Each of the $10,000$ images were repeated three times in random order, giving a total of $30,000$ readings per participant. Similar to prior studies (Scotti et al., 2024a), we compute the mean of the fMRI readings for all the three occurrences and obtain a one-to-one mapping between the image and the corresponding fMRI reading. The images belong to 12 categories: animals, accessories, appliances, electronics, food, furniture, indoor, kitchen, outdoor, person, sports, and vehicles.

The dataset is already pre-processed and a brain mask for each subject is also provided to obtain the activation voxels. Similar to prior studies (Scotti et al., 2024a;b), we use preprocessed flattened fMRI voxels in 1.8-mm native volume space corresponding to the "nsdgeneral" brain region, defined by the NSD authors as the subset of voxels in posterior cortex most responsive to the visual stimuli presented (between 13,000 to 16,000 voxels per participant). We perform the ROI (region of interest) analysis for the NSD dataset considering the following five visual processing regions: body-selective regions (floc-bodies), face-selective regions (floc-faces) and scene-selective regions (floc-places), word-selective regions (floc-words), and pRF-Visual ROIs (also called Retinotopic Early Visual Cortex) (Allen et al., 2022). Note that floc-bodies, floc-faces, floc-places and floc-words are high-level visual areas while pRF-Visual ROIs are early visual areas. We list the detailed sub-ROIs of these ROIs in Appendix B.

**Estimating dataset cross-subject prediction accuracy.** To account for the intrinsic noise in biological measurements, we adapt Schrimpf et al. (2021)'s method to estimate the cross-subject prediction accuracy for a model's performance. By subsampling fMRI datasets from 4 participants, we generate all possible combinations of $s$ participants ($s \in [2,4]$) watching natural scenes, and use a voxel-wise encoding model (see Sec. 5) to predict one participant's response from others. Note that the estimated cross-subject prediction accuracy is based on the assumption of a perfect model, which might differ from real-world scenarios, yet offers valuable insights into model's performance. We present the average cross-subject prediction accuracy across voxels for the *NSD* dataset in Appendix C Fig. 8. The figure suggests that the cross-subject prediction accuracy is consistent across subjects, indicating that all subjects share a similar amount of explainable variance. Cross-subject prediction accuracy for each participant brainmaps are reported in Appendix C in Fig. 9.

## 4 METHODOLOGY

**Instruction-tuned Multimodal large language models.** To investigate whether multimodal instruction-tuned models prompted using natural language-guided instructions perfectly align with the way humans process visual information in the brain, we consider three popular modern instruction-tuned multimodal models publicly available on Huggingface (Wolf et al., 2020): InstructBLIP (Dai et al., 2023), mPLUG-Owl (Ye et al., 2023) and IDEFICS (Laurençon et al., 2023).

**InstructBLIP** (Dai et al., 2023) is a vision-language instruction-tuned model built upon the pretrained BLIP-2 model (Li et al., 2023). **mPLUG-Owl** (Ye et al., 2023) is an MLLM designed to perceive and integrate multiple modalities (visual and language) while considering visual context and information to generate corresponding outputs. **IDEFICS** (Laurençon et al., 2023) is an MLLM based on Flamingo (Zhu et al., 2024), which accepts arbitrary sequences of image and text inputs and generates text tokens. All MLLMs consist of 32 layers and produce 4096-dimensional representations at each layer. We provide more details, including model-parameters and training dataset details in Table 2 in Appendix D.

**Natural instructions.** To ensure the diversity of task-specific instructions while considering image as input, we consider 10 instructions, and extract the language-guided representations from multimodal instruction-tuned models. As shown in Table 1, the 10 natural instructions cover 6 task categories, including image captioning, visual question answering, visual relationships, commonsense reasoning, image understanding and scene recognition. These set of 10 instructions are inspired from the list of 62 multimodal tasks defined in MultiInstruct (Xu et al., 2023). We borrowed those

Table 1: Instructions for various multimodal tasks (ordered by complexity, from least to most complex.)

| Task | Description |
|---|---|
| Image Understanding | IU1: Describe the most dominant color in the image |
| | IU2: List any food items visible. |
| | IU3: How many animals are there in the image? |
| Visual Question Answering | VQ1: What is in this image? |
| | VQ2: Are there any people in this image? If yes, describe them. |
| | VQ3: What is the foreground of the image? What is in the background? |
| Image Captioning | IC: Generate some text to describe the image |
| Scene Recognition | SR: Highlight the area that shows a natural outdoor scene. |
| Commonsense Reasoning | CR: What type of environment is shown in the image? |
| Visual Relationship | VR: What kind of interaction is happening between the animate and inanimate objects here? |

tasks which are generally applicable to any image regardless of the contents in the image. We provide a sample of generated outputs for the three MLLMs across 10 instructions in Tables 3, 4 and 5 in Appendix E.

**Extraction of features from instruction-tuned multimodal models.** To extract instruction-specific representations from multimodal instruction-tuned models for the brain encoding task, we input an image and task instruction to obtain the embeddings for the language-guided instruction. We perform zero-shot inference on these models. We use the *model.generate* option because the hidden states are influenced by both the input tokens and the generated tokens, making them dependent on the generation context. The hidden states are dynamic and continuously evolve until the model generates the final token, which involves multiple forward passes in the process. For all multimodal instruction-tuned models, we use the pretrained Transformer weights, which generate hidden state representations at each layer. We then average these hidden state representations of the output generated tokens to obtain the final embedding for each image with respect to the task instruction. For uniformity of feature dimensions, we applied PCA to reduce the dimensions to 1024.

**Unimodal and multimodal models.** As a baseline for comparison, we also included a unimodal model ViT-H (Dosovitskiy et al., 2020) and the multimodal model CLIP (Radford et al., 2021a), which is an image-text alignment model, in our experiments. The pretrained ViT-H model outputs image representations from different layers, providing a 1024-dimensional feature vector across 32 encoder layers. For the CLIP model, we input both image and ground truth caption pairs and extracted 1024-dimensional representations from the CLIP-Text model.

## 5 EXPERIMENTAL SETUP

**Voxelwise encoding model.** We estimate the brain alignment of multimodal and unimodal models of a image stimulus via training standard voxel-wise encoding models (Deniz et al., 2019; Toneva & Wehbe, 2019; Schrimpf et al., 2021). We train bootstrap ridge regression based voxel-wise encoding models (Deniz et al., 2019) to predict the fMRI brain activity associated with the stimulus representations obtained from the multimodal instruction-tuned models as well as other models. Formally, for each subject, we encode the stimuli as $X_i \in \mathbb{R}^D$ and brain region voxels $Y_i \in \mathbb{R}^V$, where $D$ denotes the dimension of the image representations, and $V$ denotes the number of voxels. Overall, with $N$ such training images, we obtain $N$ training examples.

**Train-Test dataset split.** We built an encoding model for each subject as follows: We used all data samples from 9,000 natural images (unique to each subject) for training and tested generalization on samples from the test set of 1,000 images (common across all subjects). Overall, we created per-subject, per-voxel encoding models. Model details and hyper-parameter settings are in Appendix D.

**Evaluation metrics.** We evaluate our models using Pearson Correlation (PC) which is a standard metric for evaluating brain alignment (Jain & Huth, 2018; Schrimpf et al., 2021; Goldstein et al., 2022). Let $N_{te}$ be the number of images in the test set. Let $Y = \{Y_i\}_{i=1}^{N_{te}}$ and $\hat{Y} = \{\hat{Y}_i\}_{i=1}^{N_{te}}$ denote the actual and predicted value vectors for a single voxel. Thus, $Y$ and $\hat{Y} \in \mathbb{R}^{N_{te}}$. We use Pearson Correlation (PC) which is computed as $corr(Y, \hat{Y})$ where corr is the correlation function. The final measure of a model's performance is obtained by calculating Pearson correlation between the model's predictions and neural recordings. This correlation is then divided by the estimated cross-subject prediction accuracy and averaged across voxels, regions, and participants, resulting in

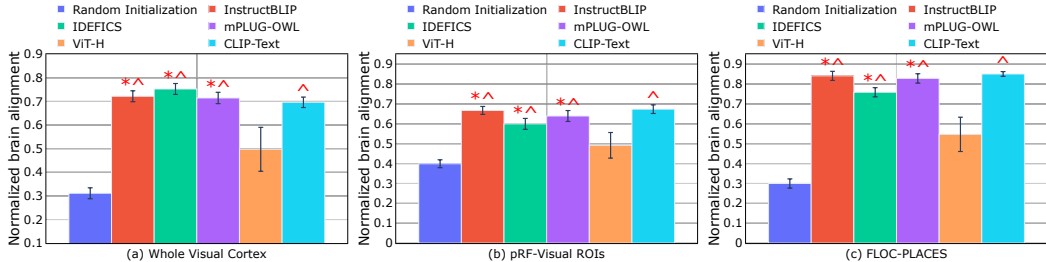

Figure 2: Whole visual cortex and ROI-based normalized brain alignment was computed by averaging across participants, layers, and voxels. Blue: Average across random initialization of the 3 MLLMs. Note that CLIP-text model uses golden oracle captions while instruct models use predicted model generations. $*$ indicates cases where MLLM embeddings are statistically significantly better than randomly initialized models, i.e., $p \leq 0.05$. $\wedge$ indicates cases where MLLMs are significantly better than unimodal vision models (ViT-H), i.e., $p \leq 0.05$. Other brain ROI plots are reported in Fig. 10 in Appendix F.

a standardized measure of performance referred to as normalized brain alignment. For calculating normalized alignment, we select the voxels whose cross-subject prediction accuracy is $\geq 0.05$.

**Variance partitioning.** To disentangle task-specific instruction representations from multimodal instruction-tuned models, we used a variance partitioning approach (de Heer et al., 2017; LeBel et al., 2021). This method measures the overlap in brain variance explained by different task-specific instruction representations. Specifically, variance partitioning separates the brain response variance that can be attributed to two models based on their unique and overlapping contributions (Vaidya et al., 2022; Deniz et al., 2019). To perform this, for every pair of instruction representations, we fit separate encoding models for each space as well as a joint encoding model, obtained by concatenating the features. Using set arithmetic, we can then derive the size of the intersection $(NBA)_v^{1 \cap 2} = (NBA)_v^1 + (NBA)_v^2 - (NBA)_v^{1 \cup 2}$, where NBA refers to normalized brain alignment, $v$ refers to a specific voxel, $(NBA)_v^1$ denotes alignment of model 1, $(NBA)_v^2$ denotes alignment of model 2 and $(NBA)_v^{1 \cup 2}$ denotes alignment of the joint model. Similarly, the unique contribution of model 1's feature space is computed as $(NBA)_v^{1 \setminus 2} = (NBA)_v^1 - (NBA)_v^{1 \cap 2}$.

## 6 RESULTS

### 6.1 MLLM REPRESENTATIONS ALIGN WELL TO HUMAN BRAIN ACTIVITY

First, we examine the brain alignment by measuring the similarity of degree of brain predictivity using representations extracted from multimodal instruction-tuned models (MLLMs), focusing on both whole visual cortex and specific visual function localizers. For each MLLM, we calculate the average normalized brain alignment across 10 instructions, multiple subjects, and different MLLM layers, using the NSD dataset. Additionally, we report baseline performance using randomly initialized versions of the InstructBLIP, mPLUG-OWL and IDEFICS models. We further compare the brain alignment performance of these MLLMs with unimodal vision model (ViT-H) and multimodal CLIP-Text model.

**Whole visual cortex analysis.** Fig. 2 (a) presents the average normalized brain alignment for whole-brain analysis across six different settings. The results demonstrate that MLLMs significantly outperform randomly initialized models in terms of brain alignment. Moreover, we find that multimodal instruction-tuning improves brain alignment over unimodal ViT-H models. Notably, the superior performance of pretrained MLLMs compared to randomly initialized models indicates that natural guided instructions yield more brain-relevant representations, leading to greater alignment with brain activity. Additionally, MLLMs perform on par with, or better than, the CLIP-Text model, despite the latter using ground-truth captions, whereas MLLMs (InstructBLIP, mPLUG-OWL and IDEFICS) use mean pooling over predicted output tokens based on natural instructions.

**ROI analysis.** We further examine how instruction-tuning enhances the alignment of MLLM representations with brain activity, focusing on specific brain regions. To do this, we compute the normalized brain alignment separately across five visual functional localizers, as discussed in Section 3. Fig. 2 (b) and (c) show the normalized alignment across six settings for pRF-Visual ROIs and

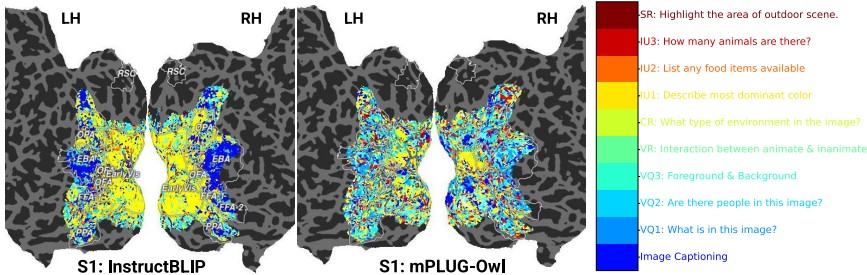

Figure 3: Each voxel is color coded with the instruction (out of 10) that led to the highest normalized brain alignment. The color bar highlights color codes for each instruction. The voxels are projected onto the flattened cortical surface of a representative subject (subject S1) for two MLLMs. Similar brain maps for other subjects are in Appendix G.

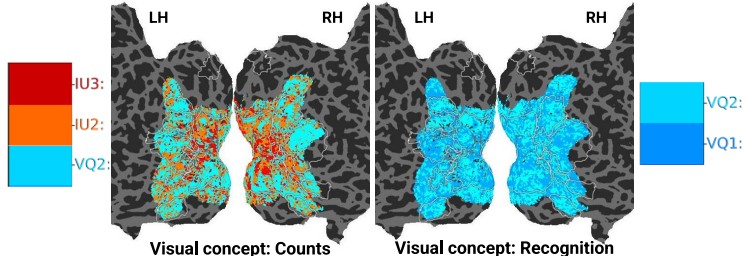

Figure 4: Voxels specific to visual concepts groups: Counts *(*Left) and Recognition *(*Right). The color bar from Fig. 3 highlights color codes for each instruction. The voxels are projected onto the flattened cortical surface of a representative subject (subject S1).

FLOC-PLACES ROIs (see additional figures in Appendix F). From Fig. 2 (b) and (c), we make the following observations: (1) Similar to the whole visual cortex analysis, MLLMs exhibit significantly better brain alignment compared to baseline (randomly initialized) and unimodal ViT-H models. (2) Interestingly, baseline performance is closer to that of unimodal models in early visual ROIs (i.e. pRF-Visual ROIs), whereas the difference becomes more pronounced in floc-place regions. This trend is also observed in MLLMs, where the normalized alignment reaches ∼0.8 in place regions but drops to ∼0.6 in early visual ROIs, suggesting that high-level visual areas provide better brain-relevant representations in MLLMs than early visual areas. This finding agrees with previous work which suggests that multimodal models better predicts neural responses in the high-level visual cortex than previous vision-only models like CNNs (Wang et al., 2022). Similar patterns are observed in other high-level visual areas (faces, bodies, words), as detailed in Appendix F.

## 6.2 ENCODING PERFORMANCES OF TASK-SPECIFIC INSTRUCTIONS FROM MLLMS

While MLLM representations at both whole visual cortex level and in visual functional ROIs demonstrate that multimodal instruction-tuned models improve brain alignment for task-specific instructions, we are also interested in understanding the importance of each task instruction and conducting a layer-wise analysis to examine the trends in brain alignment across different models.

**Which task-specific instructions are highly correlated to visual function localizers?** To investigate which instructions are more effective in predicting particular visual functional localizers, we analyze the voxels as follows. For each voxel, we select the instruction that results in the highest normalized brain alignment and apply the instruction-specific color code to the voxel. The color-scheme corresponding to each instruction is reported in Fig. 3. The figure also displays brain maps for the InstructBLIP, and mPLUG-OWL for Subject 1, where the voxel color codes are projected onto the flattened cortical surface of the representative subject. Similar brain maps for other subjects are in Appendix G.

From Fig. 3, we make the following observations: (i) Representations related to image captioning show higher brain alignment in the EBA, PPA, and FFA regions for both InstructBLIP and IDEFICS models, while the mPLUG-OWL model demonstrates higher alignment mainly in the EBA region. (ii) Instructions related to image understanding (e.g., "describe the most dominant color") result in higher brain alignment in early visual regions for both the InstructBLIP and mPLUG-OWL models,

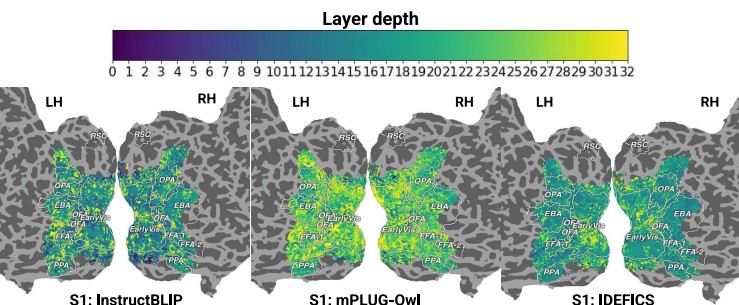

Figure 5: Each voxel is color coded with the MLLM layer number (out of 33) that led to the highest normalized brain alignment. The color bar highlights color codes for each layer. The voxels are projected onto the flattened cortical surface of a representative subject (subject S1) for three MLLMs. Similar brain maps for other subjects are in Appendix H.

while IDEFICS shows only marginal alignment with this instruction. (iii) Visual question-answering instructions, such as "are there people in this image?" and "foreground and background of the image," contribute significantly to brain alignment in higher visual regions, particularly in the PPA and FFA regions. (iv) In contrast to the above three instructions, other instructions (such as "list any food items available" and "how many animals are there?") result into some voxels with high alignment predictions, while the remaining instructions do not result in voxels with significant alignment.

In order to draw some more insights of why some instructions might be more pronounced than other in encoding the brain activity, we clustered instructions into five visual concepts - 1) Count 2) Recognition 3) Color/Texture 4) Positional Understanding and 5) General Scene Understanding. This is further motivated by our interest in uncovering which brain regions are responsible for different visual concepts.

**Do task-specific instructions from MLLMs account for visual concepts understanding?** To examine whether task-specific instructions from MLLMs capture different visual concepts and to understand how these concepts are processed in the brain's visual regions, we group the instructions into five categories: (i) *Count* (counting objects, animals, people within the image) - IU2, IU3, VQ2; (ii) *Recognition* (recognizing objects, animals, or people) - VQ1, VQ2; (iii) *Color/Texture* - IU1; (iv) *Positional Understanding* (foreground/background, right/left, upper/bottom positions) - VQ3; and (v) *General Scene Understanding* - CR.

Similar to task-specific instructions, we analyze the voxels from visual concept groups as follows. We use the same color scheme corresponding to each instruction as reported in Fig. 3. The Fig. 4 displays brain maps for the InstructBLIP model for Subject 1, with voxel color codes projected onto the flattened cortical surface of the representative subject for two concept groups: Count and Recognition. We make the following observations: (i) For the visual concept *Count*, the VQ2 instruction results in higher brain alignment in high-level visual regions, while IU2 and IU3 instructions show higher alignment in early visual regions. This suggests that MLLMs capture count-related visual concepts effectively across instructions, showing alignment with brain activity. (ii) For *Recognition*, both VQ1 and VQ2 instructions exhibit distributed brain alignment across both high-level visual regions and early visual regions. (iii) In contrast to *Count* and *Recognition*, for visual concepts like *Color*, *Positional Understanding*, and *Scene Understanding*, MLLMs display similar brain alignment patterns, irrespective of the specific visual concept. Therefore, further improvements may be needed for MLLMs to achieve better specificity in processing a broader range of visual concepts. Similar brain maps for other subjects are in Appendix G.

**Which layers of MLLMs are highly correlated to visual function localizers?** To explore whether the effectiveness of layer-wise representations from MLLMs varies in relation to visual functional localizers, we analyze the voxels as follows. For each voxel, we select the layer that results in the highest normalized brain alignment and apply a color code for the 33 layers across the three MLLMs. Fig. 5 presents brain maps for the InstructBLIP, mPLUG-Owl and IDEFICS, where the voxels with their corresponding color codes are projected onto the flattened cortical surface of the representative subject (Subject 1). Similar brain maps for other subjects are in Appendix H. From Fig. 5, we make the following observations: (i) Across all functional localizers, the middle layers of the InstructBLIP and IDEFICS models show greater brain alignment for higher visual regions, whereas the later layers

| | IC | VQ1 | VQ2 | VQ3 | VR | CR | IU1 | IU2 | IU3 | SR |
|---|---|---|---|---|---|---|---|---|---|---|
| IC | | 0.446 | 0.399 | 0.371 | 0.363 | 0.417 | 0.392 | 0.341 | 0.363 | 0.369 |
| VQ1 | 0.446 | | 0.410 | 0.363 | 0.352 | 0.432 | 0.380 | 0.338 | 0.356 | 0.355 |
| VQ2 | 0.399 | 0.410 | | 0.338 | 0.335 | 0.386 | 0.343 | 0.305 | 0.317 | 0.326 |
| VQ3 | 0.371 | 0.363 | 0.338 | | 0.315 | 0.347 | 0.334 | 0.293 | 0.314 | 0.309 |
| VR | 0.363 | 0.352 | 0.335 | 0.315 | | 0.362 | 0.325 | 0.290 | 0.313 | 0.301 |
| CR | 0.417 | 0.432 | 0.386 | 0.347 | 0.362 | | 0.370 | 0.329 | 0.363 | 0.339 |
| IU1 | 0.392 | 0.380 | 0.343 | 0.334 | 0.325 | 0.370 | | 0.289 | 0.339 | 0.336 |
| IU2 | 0.341 | 0.338 | 0.305 | 0.293 | 0.290 | 0.329 | 0.289 | | 0.296 | 0.299 |
| IU3 | 0.363 | 0.356 | 0.317 | 0.314 | 0.313 | 0.363 | 0.339 | 0.296 | | 0.298 |
| SR | 0.369 | 0.355 | 0.326 | 0.309 | 0.301 | 0.339 | 0.336 | 0.299 | 0.298 | |

Figure 6: Shared explained variance between pairs of task-specific instructions. A higher shared variance indicates that both instructions share similar features, resulting in greater shared explained variance in the brain. The diagonal cells are empty since the shared variance between pairs of the same task is identical.

are more aligned with early visual regions. In contrast, the later layers of the mPLUG-Owl model result in higher brain alignment for both higher and early visual regions. This variation across the 3 MLLMs may be due to the difference in the underlying language decoder models, which generate output tokens, capture contextual representations, and influence the alignment trend across the layers. This finding is consistent with studies on brain alignment in language models, which have shown that middle layers of these models tend to exhibit higher brain alignment (Toneva & Wehbe, 2019; Caucheteux & King, 2020). (ii) Unlike the mPLUG-Owl model, both the InstructBLIP and IDEFICS models show minimal impact of later layer alignment on high-level visual regions.

### 6.3 PARTITIONING EXPLAINED VARIANCE BETWEEN TASK-SPECIFIC INSTRUCTIONS

While the previous analysis reveals that not all instructions result in higher brain alignment across visual ROIs, we disentangle representations of task-specific instructions to measure the overlap in brain variance explained by MLLMs. To accomplish this we use variance partitioning approach discussed in Section 5. Using this approach, we measure the brain response variance explained by pairs of instruction representations, separating their unique and overlapping contributions. Variance partitioning quickly becomes intractable as the number of feature spaces increases. Thus we restrict our analysis to pairwise comparisons that involve layers with max normalized brain alignment.

**Shared variance across task-specific instructions.** Fig. 6 presents the shared variance between task prompts for the InstructBLIP model, averaged across all subjects. We observe the following from this figure: (i) High Overlap of Image Captioning (IC) with other prompts: The IC instruction exhibits a high degree of shared information with prompts VQ1, VQ2, CR, and IU1. The prominent brain alignment of IC across multiple prompts suggests that it acts as the general or umbrella category for the other instruction prompts. This is intuitive since in order to generate an image caption, the responses to other instructions are automatically taken into account. This broad relevance may account for its ability to capture various visual elements that are crucial across different tasks, driving consistent brain alignment. (ii) Lower shared variance with IU2 and SR: Instructions like IU2 and SR demonstrate consistently lower shared variance with other prompts, particularly VQ and IC. This indicates that these instructions may elicit distinct neural responses or emphasize specific aspects of visual stimuli not represented by the broader visual tasks linked to VQ or IC. Further, to verify these results, we plotted the percentage overlap across image categories and show it in Fig. 15. Based on that, we see that the instruction IU2 which is concerned with "food" category would naturally have low shared variance with other instructions, since the overlap of "food" category is much lower with other categories, except for "kitchen", "furniture", and "person". And, we see similar trends for the instruction SR, which explains its low shared variance, by looking at the "outdoor" and "indoor" image categories.

Overall, these observations underscore why IC, IU2, and VQ2 have task-specific brain alignments, exhibiting unique aspects of brain responses to visual regions, as illustrated in Fig. 3.

**ROI Analysis: Shared and Unique variance across task-specific instructions.**

Fig. 7 presents the unique and shared variance between task prompts—Image Captioning (IC) and Image Understanding 2 (IU2)—for the InstructBLIP model, focusing on representative subject-1. We present similar analysis between Image Captioning (IC) and Visual Question Answering 1 (VQ1) in Appendix Fig. 14. From Fig. 7, we observe the following: (i) Between IC and IU2, there is no unique variance for IU2 in the EBA region, while IC retains some unique variance. Additionally, other high-level visual regions show a similar percentage of unique and shared variance for both IC and IU2 instructions. (ii) In contrast to IC and IU2, between IC and VQ1, there is no unique

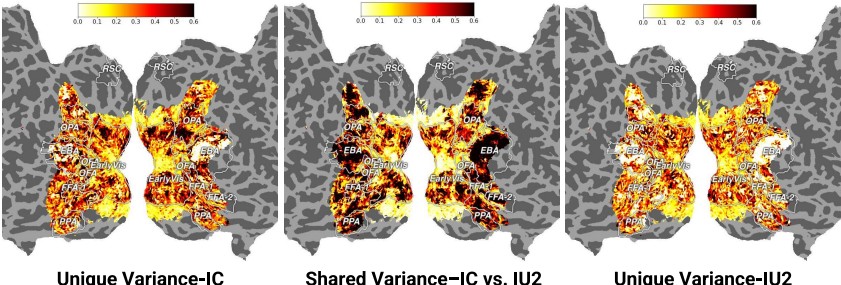

**Unique Variance-IC**    **Shared Variance−IC vs. IU2**    **Unique Variance-IU2**

Figure 7: Unique & shared variances between pairs of task-specific instructions for the InstructBLIP model, focusing on representative subject-1. Image Captioning (IC) vs. Image Understanding (IU2). Fig. 14 in Appendix I shows similar plots for IC vs. Visual Question Answering 1 (VQ1).

variance in the high-level visual region EBA, as this region is explained by shared information between the models. (iii) In other high-level visual regions, a portion of the variance is unique to each model, though the majority is explained by shared variance. Overall, these findings highlight the role of shared neural processes across task-specific instructions in high-level visual regions while also demonstrating that different task instructions can drive distinct neural responses in specific regions. However, the fact that the majority of variance is shared between instructions suggests room for improvement in the MLLM models. Enhancing the models' ability to capture more task-specific information could lead to greater precision in predicting brain responses and better differentiation between various types of instructions. More detailed shared and unique variance analysis across task-specific instructions for whole visual cortex and ROI level are reported in Appendix L. Our findings demonstrate that shared variance increases from early visual to higher visual areas, reflecting the hierarchical nature of visual processing.

## 7    DISCUSSION AND CONCLUSION

Using instruction-tuned representations from MLLMs for various instructions, we evaluated how well these representations predict fMRI brain activity when participants viewed naturalistic image stimuli. Additionally, we compared different MLLMs' representations, assessing their alignment with each instruction across five visual brain regions. We show that MLLMs exhibit significantly better brain alignment than vision-only models and perform comparably to multimodal models.

Our analysis of instruction-tuned MLLMs and their brain alignment reveals several key conclusions: (1) The effectiveness of task-specific instructions in predicting visual brain activity across different regions reveals that, although all three MLLMs generate task-specific output tokens based on instructions, not all instructions lead to increased brain alignment across all regions, as expected. Specifically, certain instructions (IC, VQ2, and IU1) may be more effective than others in encoding brain activity. This suggests that these instructions might serve as general or umbrella categories for other instruction prompts. This is further explained through a variance partitioning approach. (2) To uncover which brain regions are responsible for visual concept understanding, we examine how different instructions from MLLMs capture various visual concepts, such as counts and recognition, as well as other concepts like color, positional understanding, and general scene understanding. We find that while MLLMs effectively capture count-related and recognition-related visual concepts across instructions, they exhibit similar brain alignment patterns for other concepts. (3) By employing a variance partitioning approach, we measure the brain response variance explained by pairs of instruction representations, distinguishing between their unique and overlapping contributions. We find that most of the variance is shared across instructions, with a high overlap between Image Captioning (IC) and other prompts but lower overlap with image understanding and scene recognition. These results highlight a potential area for improvement in MLLMs, particularly in achieving better differentiation between various types of instructions. (4) The layer-wise representations from MLLMs correlate with visual functional localizers. In examining layer-wise trends in brain alignment, we find that both InstructBLIP and IDEFICS demonstrate better brain alignment in middle layers for higher visual regions, while later layers align more with early visual regions. In contrast, mPLUG-Owl achieves higher brain alignment in the later layers for both high-level and early visual brain regions. This highlights the differences in information processing across the layers of MLLMs. We make the code publicly available[1]. Lastly, we discuss limitations of our work in Appendix N.

## 8 ETHICS STATEMENT

We did not create any new neural recordings data as part of this work. We used the NSD dataset which is publicly available without any restrictions. NSD dataset can be downloaded from `https://naturalscenesdataset.org/`. Please read their terms of use[2] for more details.

We do not foresee any harmful uses of this technology.

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

## A  Overview of Appendix Sections

- Section B: Visual functional localizers
- Section C: Cross-subject brain predictivity
- Section D Details of MLLMs with training details and their parameters
- Section E: Model generated outputs across instructions
- Section F: Normalized brain alignment: Visual functional localizers
- Section G: Brain Maps for Task-specific Instructions
- Section H: Brain Maps for different MLLM Layers
- Section I: ROI Analysis: Shared and Unique variance across task-specific instructions.
- Section J: Normalized Brain Alignment across different image categories.
- Section K: Comparison of Instruction-tuned MLLMs, Non-Instruction-tuned MLLMs and text-based LLMs.
- Section L: Whole Visual Cortex and ROI Analysis: Shared and Unique variance across task-specific instructions.
- Section M: Image only / Instruction only input to the instruction-tuned MLLM.
- Section N: Limitations.

## B  Visual functional localizers

The NSD data covers five brain regions of interest (ROIs) in the human brain with the following subdivisions: early visual (pRF-Visual ROIs) and high-level visual (floc-bodies, floc-words, floc-faces and floc-places) (Allen et al., 2022).

- floc-bodies is a collection of manually drawn ROIs based on results of the floc experiment. These ROIs consist of EBA, FBA-1, FBA-2, and mTL-bodies ("mid temporal lobe bodies").
- floc-words is a collection of manually drawn ROIs based on results of the floc experiment. These ROIs consist of OWFA, VWFA-1, VWFA-2, mfs-words ("mid fusiform sulcus words"), and mTL-words ("mid temporal lobe words").
- floc-faces is a collection of manually drawn ROIs based on results of the floc experiment. These ROIs consist of OFA, FFA-1, FFA-2, mTL-faces ("mid temporal lobe faces"), and aTL-faces ("anterior temporal lobe faces").
- floc-places is a collection of manually drawn ROIs based on results of the floc experiment. These ROIs consist of OPA, PPA, and RSC.
- pRF-Visual ROIs (Retinotopic Early Visual) is a collection of manually drawn ROIs based on results of the prf experiment. These ROIs consist of V1v, V1d, V2v, V2d, V3v, V3d, and hV4.

## C  Cross-subject brain predictivity

We estimate cross-subject prediction accuracy for the NSD dataset and present the average accuracy across voxels for each subject in Fig. 8. The figure suggests that the cross-subject prediction accuracy is consistent across subjects, indicating that all subjects share a similar amount of explainable variance.

## D  Details of MLLMs with training details and their parameters

We provide details about model parameters, training dataset details, training procedure details and task-specific instructions for instruction tuning of these models in Table 2.

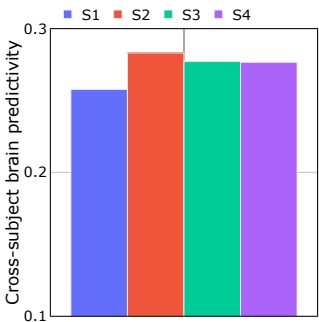

Figure 8: Cross-subject prediction accuracy for each subject of NSD dataset.

**InstructBLIP** (Dai et al., 2023) is a vision-language instruction-tuned model built upon the pre-trained BLIP-2 model (Li et al., 2023). It leverages a diverse set of instruction data (26 different datasets) to train a MLLM, which comprises an image encoder, a large language model (LLM), and a Query Transformer (Q-Former) that serves as a bridge between the two. We utilize the instructblip-vicuna-7b version, which consists of 32 layers and produces 4096-dimensional representations.

**mPLUG-Owl** (Ye et al., 2023) is an MLLM designed to perceive and integrate multiple modalities (visual and language) while considering visual context and information to generate corresponding outputs. The model is trained on a language modeling task, which involves learning to generate subsequent tokens based on the preceding context. We utilize the mplug-owl-llama-7b version, which consists of 32 layers and 4096-dimensional representations.

**IDEFICS** (Laurençon et al., 2023) is an MLLM based on Flamingo (Zhu et al., 2024), which accepts arbitrary sequences of image and text inputs and generates text tokens. We utilize the idefics-9b-instruct version (the model obtained by further training IDEFICS on supervised fine-tuning and instruction fine-tuning datasets), which consists of 32 layers and 4096-dimensional representations.

Table 2: MLLM Training Details and Parameters

| Model | Architecture | Training Dataset | Task-Specific Instructions | # Parameters | Training Procedure |
|---|---|---|---|---|---|
| InstructBLIP | Transformer-based multimodal | Large-scale image-text pairs; instruction-following datasets | Image captioning, visual question answering, image understanding | 12B | Supervised on image-text data, followed by instruction-tuning |
| mPLUG-OWL | Vision-language transformer | Image-text pairs; multimodal datasets | Visual reasoning, image captioning, text generation | 14B | Masked language modeling, next-word prediction, visual grounding, instruction-tuning |
| IDEFICS | Multimodal transformer | COCO, Visual Genome; instruction-tuning | Image captioning, visual question answering, scene understanding | 10B | Contrastive learning on image-text pairs, task-specific fine-tuning |

**Implementation details for reproducibility.** All feature extraction experiments were conducted on a machine equipped with an NVIDIA A100 GPU with 80 GB of GPU RAM, partitioned into two devices of 40 GB each. The voxelwise encoding models were trained on NVIDIA GeForce RTX 3050 GPU with 4GB of GPU RAM. We used bootstrap ridge-regression with the following parameters: MSE loss function; L2-decay ($\lambda$) varied from $10^{-1}$ to $10^3$; the best $\lambda$ was chosen by tuning on validation data that comprised a randomly chosen 10% subset from the train set used only for hyper-parameter tuning.

# E    MODEL GENERATED OUTPUTS ACROSS INSTRUCTIONS

Tables 3, 4, 5 and 6 show model generated outputs for a sample image from the NSD dataset using InstructBLIP, mPLUG-Owl, IDEFICS and BLIP-2 models respectively.

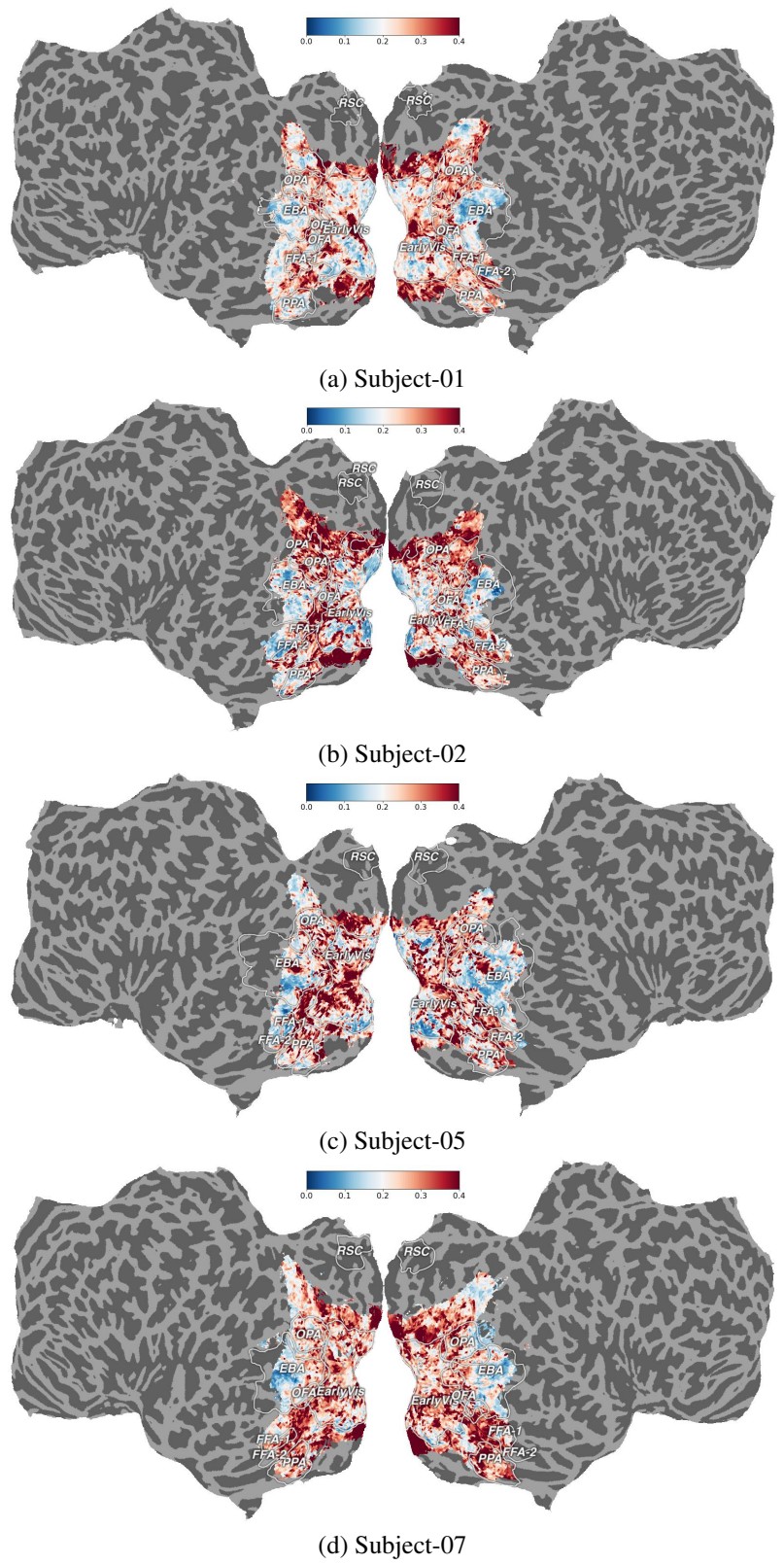

Figure 9: Contrast of estimated cross-subject prediction accuracy for the participants for the NSD dataset.

Table 3: InstructBLIP generated outputs for a sample image from the NSD dataset.

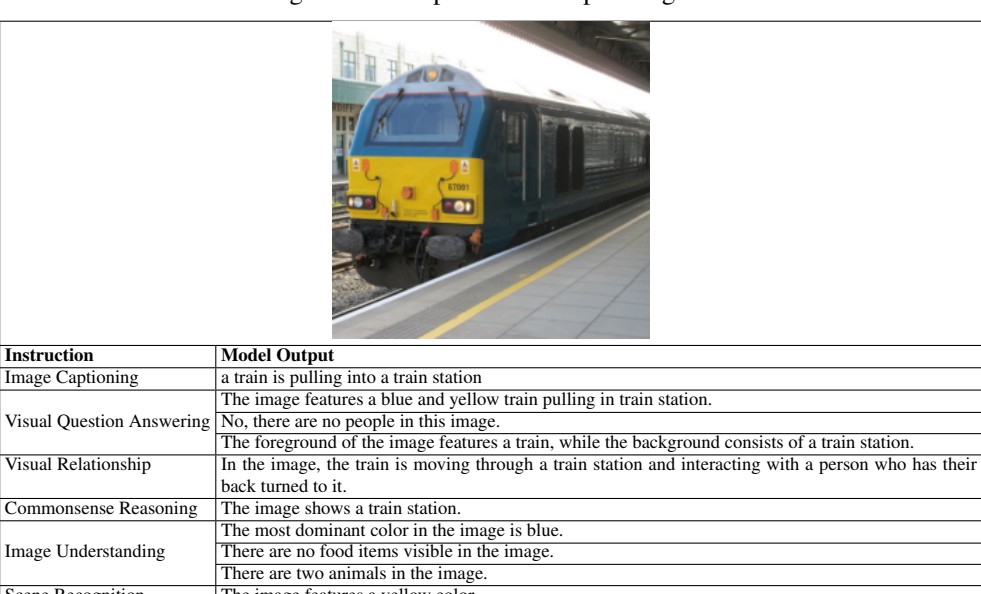

| Instruction | Model Output |
|---|---|
| Image Captioning | a train is pulling into a train station |
| Visual Question Answering | The image features a blue and yellow train pulling in train station. |
| | No, there are no people in this image. |
| | The foreground of the image features a train, while the background consists of a train station. |
| Visual Relationship | In the image, the train is moving through a train station and interacting with a person who has their back turned to it. |
| Commonsense Reasoning | The image shows a train station. |
| Image Understanding | The most dominant color in the image is blue. |
| | There are no food items visible in the image. |
| | There are two animals in the image. |
| Scene Recognition | The image features a yellow color. |

Table 4: mPLUG-Owl generated outputs for a sample image from the NSD dataset.

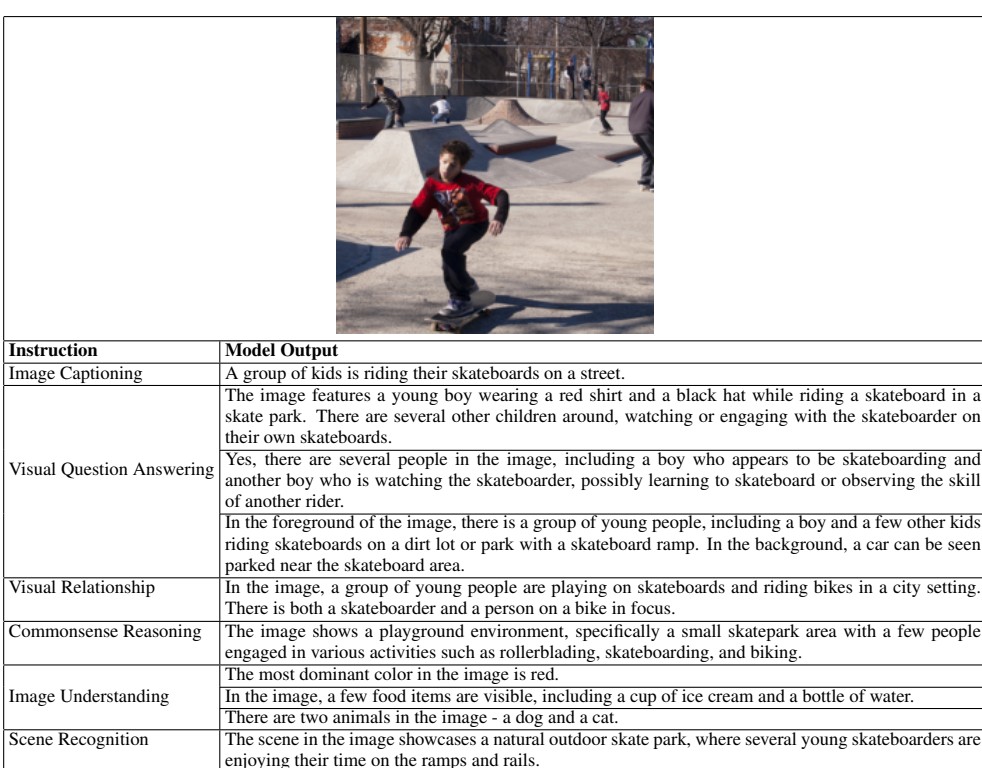

| Instruction | Model Output |
|---|---|
| Image Captioning | A group of kids is riding their skateboards on a street. |
| Visual Question Answering | The image features a young boy wearing a red shirt and a black hat while riding a skateboard in a skate park. There are several other children around, watching or engaging with the skateboarder on their own skateboards. |
| | Yes, there are several people in the image, including a boy who appears to be skateboarding and another boy who is watching the skateboarder, possibly learning to skateboard or observing the skill of another rider. |
| | In the foreground of the image, there is a group of young people, including a boy and a few other kids riding skateboards on a dirt lot or park with a skateboard ramp. In the background, a car can be seen parked near the skateboard area. |
| Visual Relationship | In the image, a group of young people are playing on skateboards and riding bikes in a city setting. There is both a skateboarder and a person on a bike in focus. |
| Commonsense Reasoning | The image shows a playground environment, specifically a small skatepark area with a few people engaged in various activities such as rollerblading, skateboarding, and biking. |
| Image Understanding | The most dominant color in the image is red. |
| | In the image, a few food items are visible, including a cup of ice cream and a bottle of water. |
| | There are two animals in the image - a dog and a cat. |
| Scene Recognition | The scene in the image showcases a natural outdoor skate park, where several young skateboarders are enjoying their time on the ramps and rails. |

## F  NORMALIZED BRAIN ALIGNMENT: VISUAL FUNCTIONAL LOCALIZERS

Fig. 10 shows ROI-based normalized brain alignment computed by averaging across participants, layers, and voxels. The figure shows the alignment values for FLOC-BODIES, FLOC-FACES and FLOC-WORDS regions.

Table 5: IDEFICS generated outputs for a sample image from the NSD dataset.

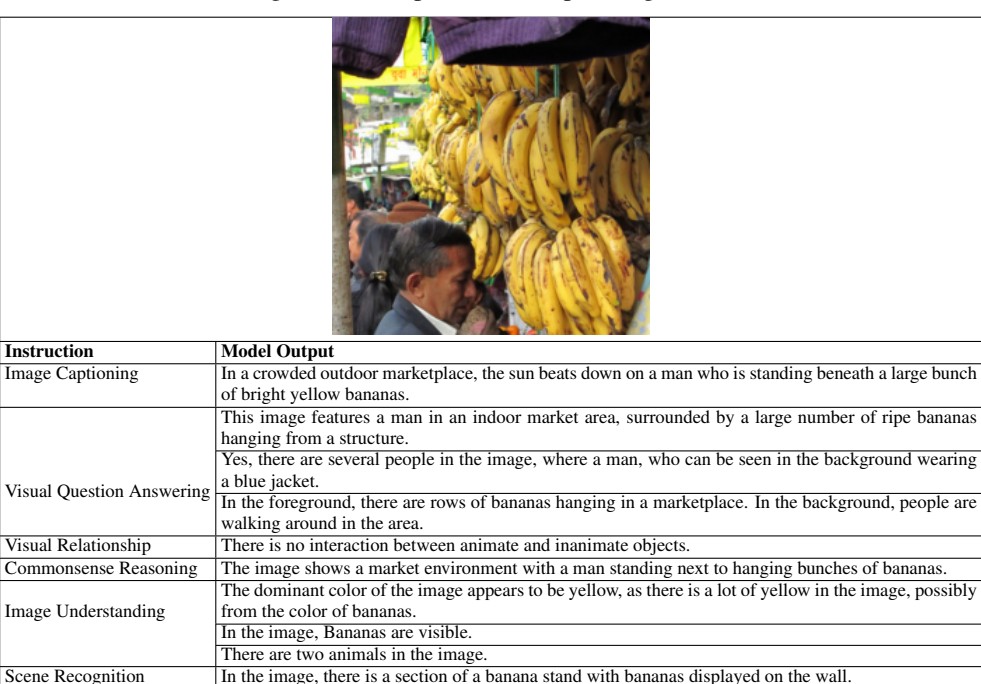

| Instruction | Model Output |
|---|---|
| Image Captioning | In a crowded outdoor marketplace, the sun beats down on a man who is standing beneath a large bunch of bright yellow bananas. |
| Visual Question Answering | This image features a man in an indoor market area, surrounded by a large number of ripe bananas hanging from a structure. |
| | Yes, there are several people in the image, where a man, who can be seen in the background wearing a blue jacket. |
| | In the foreground, there are rows of bananas hanging in a marketplace. In the background, people are walking around in the area. |
| Visual Relationship | There is no interaction between animate and inanimate objects. |
| Commonsense Reasoning | The image shows a market environment with a man standing next to hanging bunches of bananas. |
| Image Understanding | The dominant color of the image appears to be yellow, as there is a lot of yellow in the image, possibly from the color of bananas. |
| | In the image, Bananas are visible. |
| | There are two animals in the image. |
| Scene Recognition | In the image, there is a section of a banana stand with bananas displayed on the wall. |

Table 6: BLIP-2 generated outputs for a sample image from the NSD dataset.

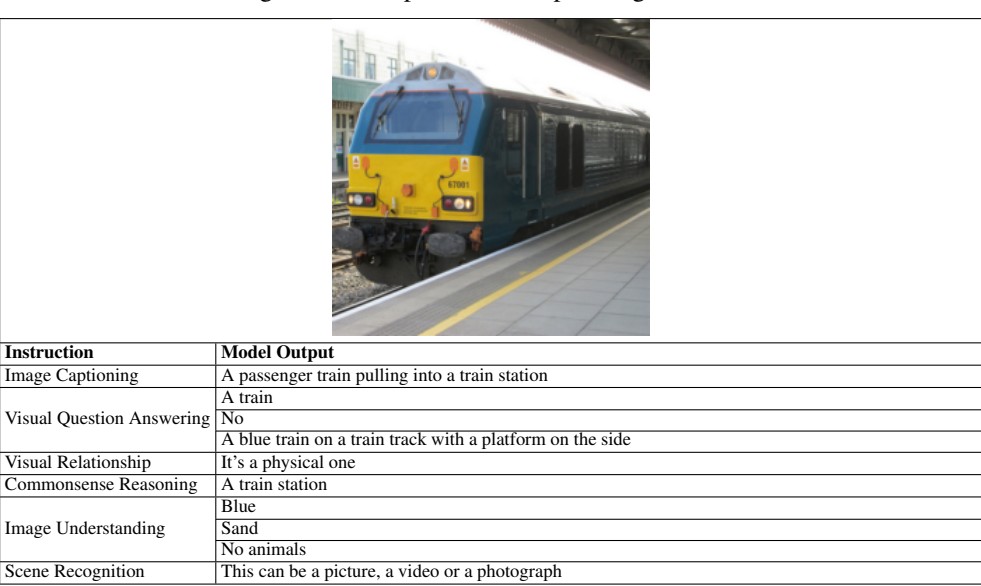

| Instruction | Model Output |
|---|---|
| Image Captioning | A passenger train pulling into a train station |
| Visual Question Answering | A train |
| | No |
| | A blue train on a train track with a platform on the side |
| Visual Relationship | It's a physical one |
| Commonsense Reasoning | A train station |
| Image Understanding | Blue |
| | Sand |
| | No animals |
| Scene Recognition | This can be a picture, a video or a photograph |

We conducted a non-parametric one-way ANOVA test to analyze the normalized brain alignment differences between early and higher visual ROIs for each instruction-tuned MLLM across four subjects.

The results for InstructBLIP Model are as follows. The one-way ANOVA test revealed that higher visual ROIs have significantly higher normalized brain alignment than early visual ROIs, with a p-value of 0.008 and an F-statistic of 14.60.

**Pairwise Comparisons Between Early Visual and Higher Visual ROIs:** We further performed ANOVA tests between early visual ROIs and each specific higher visual ROI. The results indicate that for most higher visual ROIs, the brain alignment is significantly better compared to early vi-

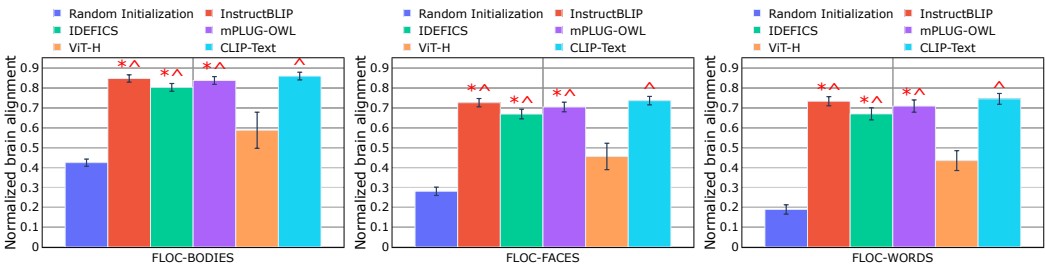

Figure 10: ROI-based normalized brain alignment was computed by averaging across participants, layers, and voxels. Blue: Average across random initialization of the 3 MLLMs. Note that CLIP-text model uses golden oracle captions while instruct models use predicted model generations. $*$ indicates cases where MLLM embeddings are statistically significantly better than randomly initialized models, i.e., $p \leq 0.05$. $\wedge$ indicates cases where MLLMs are significantly better than unimodal vision models (ViT-H), i.e., $p \leq 0.05$.

sual ROIs: Early Visual vs. FLOC-PLACES (p-value: 0.006, F-statistic: 41.45), Early Visual vs. FLOC-Bodies (p-value: 0.001, F-statistic: 32.91), Early Visual vs. FLOC-Faces (p-value: 0.006, F-statistic: 41.02), Early Visual vs. FLOC-Words (p-value: 0.14 (not statistically significant), F-statistic: 2.83). These results quantitatively confirm that multimodal models such as InstructBLIP achieve significantly better alignment in higher visual ROIs than in early visual ROIs.

We performed a similar analysis to include all instruction-tuned MLLMs and conducted a one-way ANOVA test to compare normalized brain alignment between early visual and higher visual ROIs. The one-way ANOVA test revealed that higher visual ROIs have significantly higher normalized brain alignment than early visual ROIs, with a p-value of 0.009 and an F-statistic of 13.85.

## G  BRAIN MAPS FOR TASK-SPECIFIC INSTRUCTIONS

The Fig. 11 displays brain maps for the InstructBLIP, and mPLUG-OWL for Subject 1, where the voxel color codes are projected onto the flattened cortical surface of the representative subject.

The Fig. 12 displays brain maps for the InstructBLIP model for Subject 1, with normalized brain alignment of voxels are projected onto the flattened cortical surface of the representative subject for two concept groups: Color, Position and Scene understanding.

## H  BRAIN MAPS FOR DIFFERENT MLLM LAYERS

Fig. 13 presents brain maps for the InstructBLIP, mPLUG-Owl and IDEFICS, where the voxels with their corresponding color codes are projected onto the flattened cortical surface of the representative subjects (Subjects 2, 5 and 7).

## I  ROI ANALYSIS: SHARED AND UNIQUE VARIANCE ACROSS TASK-SPECIFIC INSTRUCTIONS

Fig. 14 presents the unique and shared variance between task prompts—Image Captioning (IC) and Visual Question Answering 1 (VQ1)—for the InstructBLIP model, focusing on representative subject-1. From Fig. 14, we observe the following: (i) Between IC and VQ1, there is no unique variance in the high-level visual region EBA, as this region is explained by shared information between the models. (ii) In other high-level visual regions, a portion of the variance is unique to each model, though the majority is explained by shared variance. Overall, these findings highlight the role of shared neural processes across task-specific instructions in high-level visual regions while also demonstrating that different task instructions can drive distinct neural responses in specific regions. However, the fact that the majority of variance is shared between instructions suggests room for improvement in the MLLM models. Enhancing the models' ability to capture more task-specific

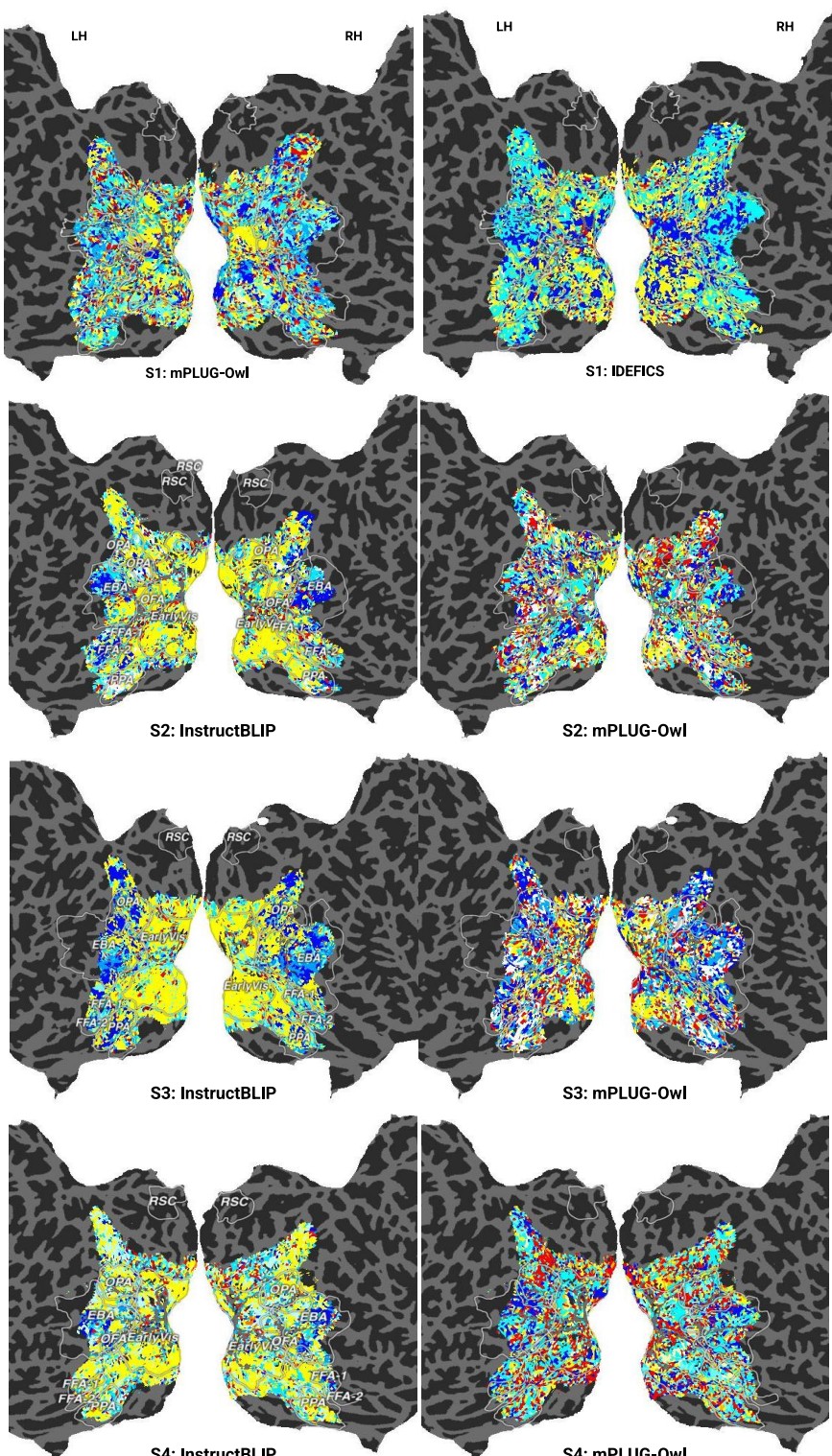

Figure 11: Each voxel is color coded with the instruction (out of 10) that led to the highest normalized brain alignment. The color bar highlights color codes for each instruction. The voxels are projected onto the flattened cortical surface of a representative subject (subject S1, S2, S5 and S7) for three MLLMs.

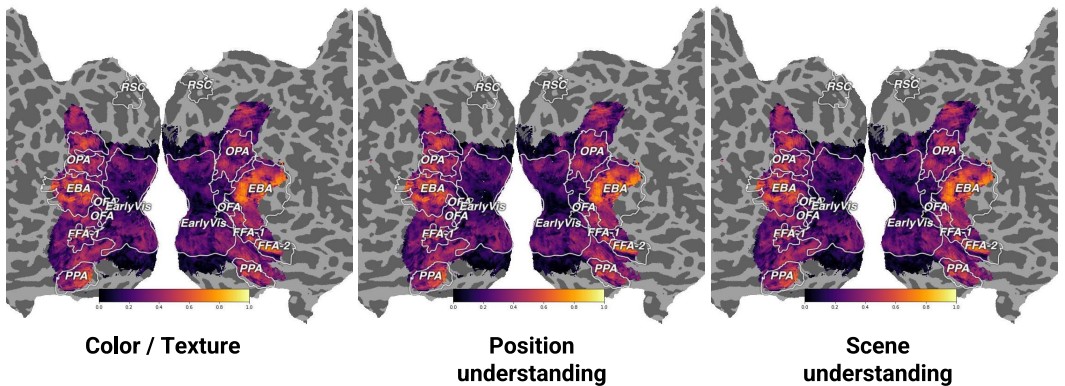

Figure 12: The visual concept specific voxels are projected onto the flattened cortical surface of a representative subject-1.

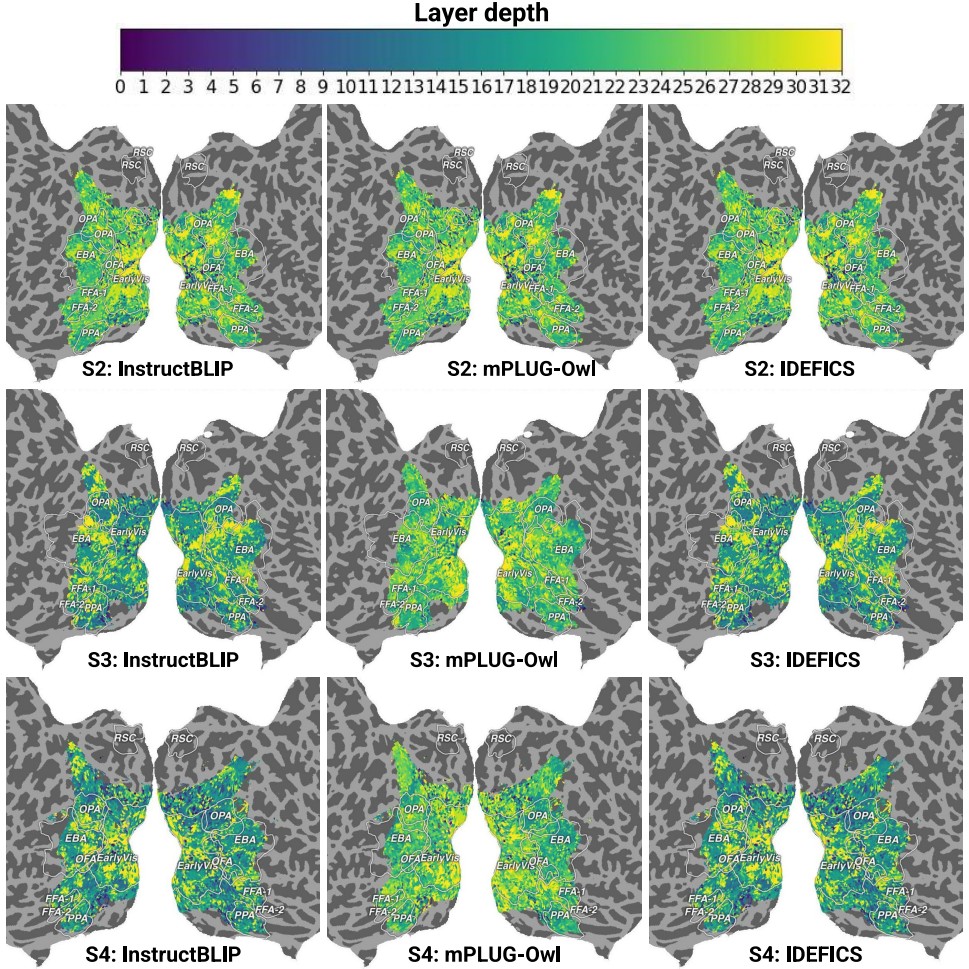

Figure 13: Each voxel is color coded with the MLLM layer number (out of 33) that led to the highest normalized brain alignment. The color bar highlights color codes for each layer. The voxels are projected onto the flattened cortical surface of a representative subject (S2, S5 and S7) for three MLLMs.

information could lead to greater precision in predicting brain responses and better differentiation between various types of instructions.

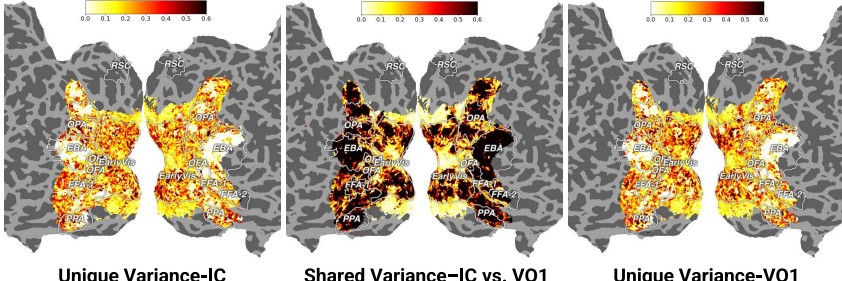

**Figure 14:** Unique and shared variances between pairs of task-specific instructions for the Instruct-BLIP model, focusing on representative subject-1. Image Captioning (IC) vs. Visual Question Answering 1 (VQ1).

## J NORMALIZED BRAIN ALIGNMENT ACROSS DIFFERENT IMAGE CATEGORIES

The NSD dataset contains images from 12 different categories. Each image can be labeled with multiple categories. Fig. 15 shows the percentage of overlap between pair of categories. Some pairs have very high overlap values as expected like (person, sports), (food, kitchen), (furniture, kitchen), (person, vehicle), etc.

To understand the effectiveness of task-specific instructions to brain alignment across various image categories, we performed a category-wise analysis, where we computed the normalized brain alignment for voxels in each category by averaging across all task instructions, and across all the 3 MLLMs. Fig. 16 shows the normalized brain alignment for five visual functional localizers: FLOC-BODIES, FLOC-PLACES, FLOC-FACES, FLOC-WORDS and pRF-Visual ROIs.

**Do the MLLM representations demonstrate better brain predictivity for certain categories of image stimuli?**

To understand the effectiveness of task-specific instructions to brain alignment across various image categories, we performed a category-wise analysis, where we computed the normalized brain alignment for voxels in each category by averaging across all task instructions, and across all the 3 MLLMs. Fig. 16 shows the normalized brain alignment for FLOC-BODIES and FLOC-PLACES ROIs. From this, we observe the following: (i) MLLMs exhibit higher normalized brain alignment for person category test images in the FLOC-BODIES region, suggesting that instruction-specific representations effectively capture body-related information, resulting in higher alignment. We also observe higher alignment in the furniture category, likely due to its 15% overlap with person-related images, as shown in Fig. 15. (ii) For FLOC-PLACES ROI, similar to FLOC-BODIES, both furniture and kitchen categories test images have higher alignment, as these categories share overlapping image content (30%). (iii) Although the appliance category has partial overlap with furniture and kitchen, MLLMs display lowest normalized brain alignment for this category. This suggests that the appliance category may contain unique features not as well captured by the MLLM's representations, or it may lack the strong shared features that drive high alignment in other categories. These trends are consistent across other visual ROIs, such as FLOC-WORDS, FLOC-FACES, and pRF-Visual ROIs, as shown in Fig. 16.

Overall, the analysis shows that MLLMs are most effective at aligning with brain activity when both task-specific and shared category features are present, suggesting that these features play a crucial role in brain alignment. However, in categories with fewer shared visual elements (e.g., appliances), alignment is weaker, indicating that these models may have difficulty fully capturing certain types of visual information.

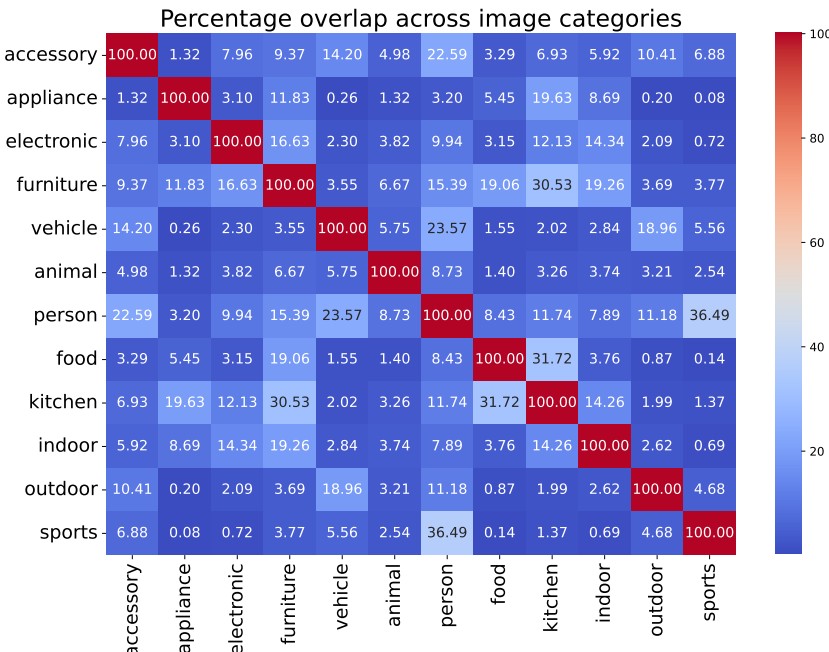

Figure 15: Percentage of overlap between pair of categories.

## K   COMPARISON OF INSTRUCTION-TUNED MLLMS, NON-INSTRUCTION-TUNED MLLMS AND TEXT-BASED LLMS

To thoroughly explore the role of instruction tuning, we conducted two additional experiments: (i) Non-instruction-tuned MLLM (BLIP-2): We passed all task instructions as input. (ii) Language Model (LLaMA-2-7B): We passed only captions as input without task instructions.

**Generated output tokens from BLIP-2.** Firstly, we present the generated output tokens from the BLIP-2 model in Table 6. This table includes examples of the generated tokens for each task instruction. We already provided predictions from InstructBLIP for the same image in Appendix Table 3. From the examples in Table 6, we observe that the generated output tokens adhere more closely to captioning instructions, regardless of the specific task instructions provided. The outputs often consist of simple responses, such as "Yes," "No," or color names, and lack detailed descriptions. In contrast, instruction-tuned MLLMs excel at providing semantically rich and conceptually grounded descriptions, as shown in Table 3, demonstrating a significant difference in their ability to follow task-specific instructions effectively.

**Comparison with normalized brain alignment scores across voxels.** Second, we present the scatter plot in Fig. 17, which compares the performance of InstructionBLIP vs. BLIP-2 for brain predictivity. The plot includes all voxels across the visual cortex with normalized brain alignment scores, where the diagonal represents identical performance for both models.

Image Captioning Task (left): The histogram shows a distribution of voxels deviating from the diagonal towards InstructBLIP, indicating that InstructBLIP performs better. However, the deviation is more pronounced for voxels with normalized brain alignment scores > 0.2.

Visual Relation Task (right): Similarly, the voxel distribution deviates significantly towards Instruct-BLIP, demonstrating its superior performance. The deviation is notably larger for this task compared to the image captioning task.

**Whole Visual Cortex vs. ROI-level Analysis** Third, we extended the analysis from Fig. 2 of the main paper by computing normalized brain alignment for the following regions: (a) Whole Visual Cortex, (b) Early Visual Cortex (pRF-Visual), and (c) High-level Visual Cortex (Bodies, Faces, Places, Words) The results in the Fig. 18, demonstrate that the non-instruction-tuned MLLM (BLIP-2) achieves brain alignment scores that are marginally below those of the CLIP-Text model.

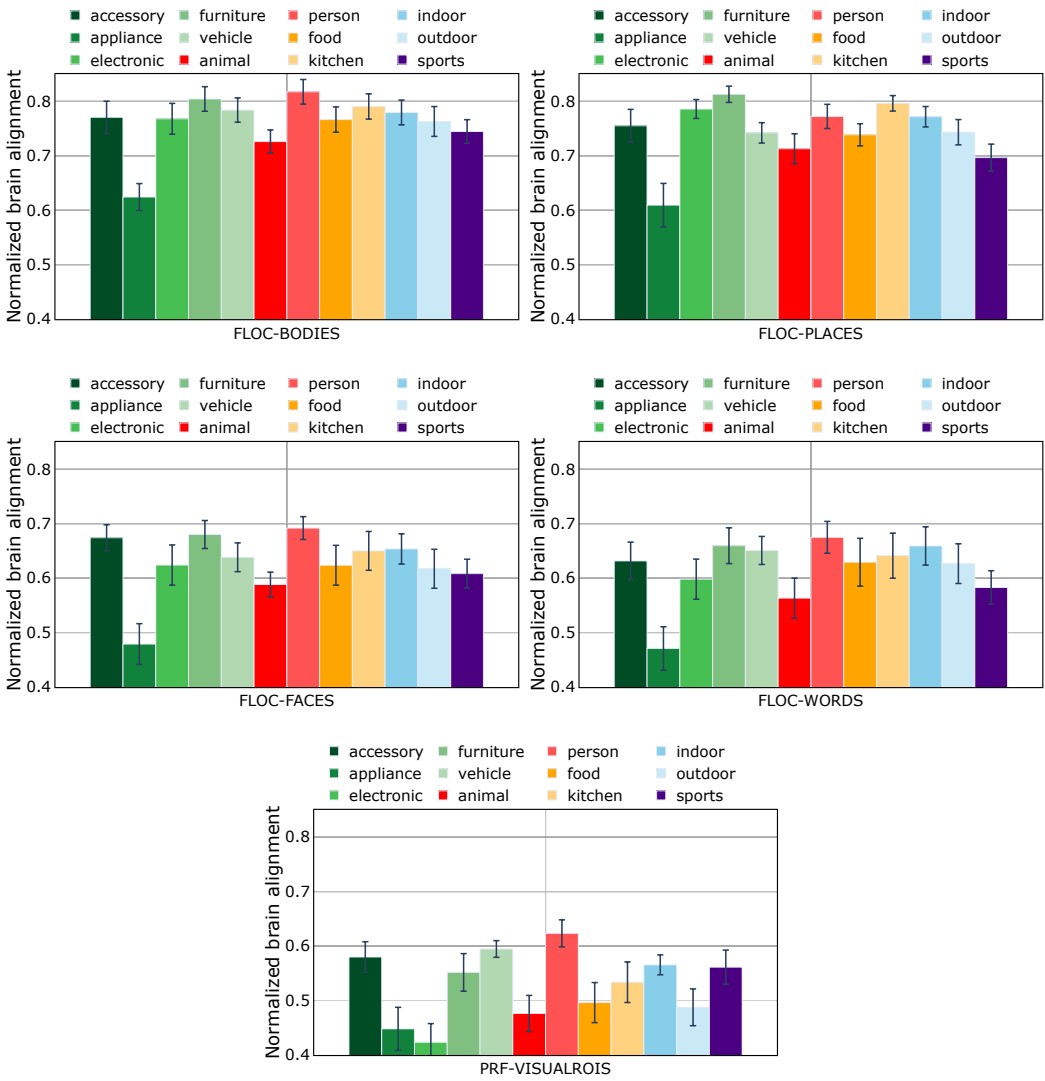

Figure 16: Average (across the 3 MLLMs) normalized brain alignment across 12 categories of images for two function localizers (FLOC-BODIES, FLOC-PLACES, FLOC-FACES, FLOC-WORDS and pRF-Visual ROIs). Error bars indicate the standard error of the mean across participants.

In contrast, the LLaMA-2-7B model demonstrates performance closer to the ViT-H model. This behavior is attributed to the fact that non-instruction-tuned models generate output tokens that adhere more closely to captioning instructions, regardless of specific task instructions.

**Key Findings**

- The performance boost in instruction-tuned MLLMs is primarily due to their ability to generate semantic descriptions and conceptually understand the elements of each scene, as opposed to merely generating task-specific answers, as seen in non-instruction-tuned MLLMs.

- Representations from the LLaMA-2-7B model, which are based solely on captions, exhibit more semantic information but lack visual understanding details. Consequently, its performance aligns more closely with visual model representations (e.g., ViT).

Overall, Instruction-tuned MLLMs demonstrate superior visual understanding when provided with task-specific instructions, as evidenced by their higher brain alignment. This reinforces the value of instruction tuning in enhancing multimodal understanding and alignment with human brain representations.

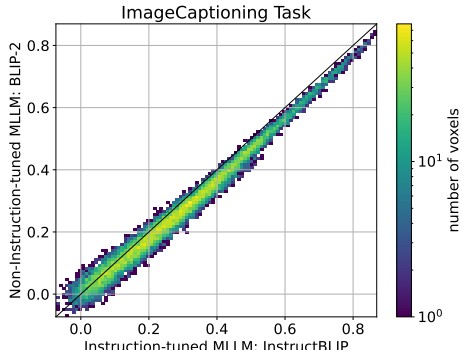 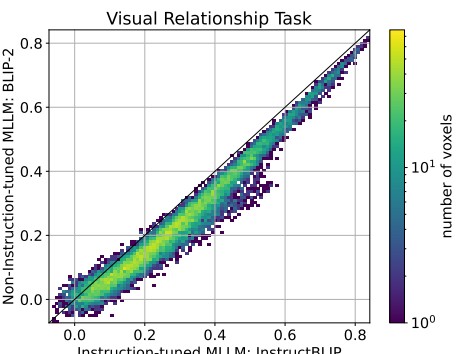

Figure 17: Comparison of MLLMs performance (Instruction-tuned vs. Non-Instruction-tuned) with a 2D scatter plot. Normalized brain alignment of all voxels is represented in the plot. The diagonal corresponds to identical performance for both models. A distribution of voxels deviating from the diagonal towards InstructBLIP means that InstructBLIP is performing better than BLIP-2. Left plot is for Image Captioning task, and right plot is for Visual Relation task. Plot shows that InstructBLIP performs better than BLIP-2.

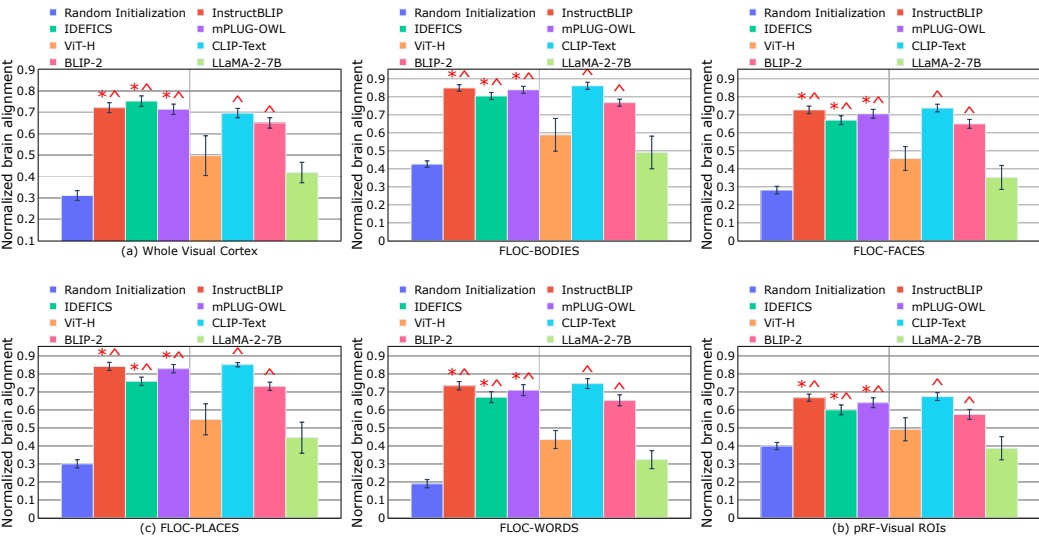

Figure 18: Whole Visual Cortex and ROI-based normalized brain alignment was computed by averaging across participants, layers, and voxels. Blue: Average across random initialization of the 3 MLLMs. Note that CLIP-text model and LLaMA models use golden oracle captions while instruct and non-instruct models use predicted model generations. ∗ indicates cases where MLLM embeddings are statistically significantly better than randomly initialized models, i.e., $p \leq 0.05$. ∧ indicates cases where MLLMs are significantly better than unimodal vision models (ViT-H), i.e., $p \leq 0.05$.

## L    WHOLE VISUAL CORTEX AND ROI ANALYSIS: SHARED AND UNIQUE VARIANCE ACROSS TASK-SPECIFIC INSTRUCTIONS

Figure 19 shows Venn diagrams representing the shared and unique variance between the IC task and nine other task-based instructions (VQ1, VQ2, VR, CR, IU1, IU2, IU3, SR) across the whole visual cortex. We make the following observations from Figure 19: (1) The higher shared variance across most task pairs (e.g., IC & VQ1, IC & CR), highlighting that tasks involving visual question answering and captioning share significant neural processing mechanisms in the visual cortex. (2) IC retains a notable amount of unique variance in most comparisons, suggesting that certain neural processes underlying image captioning are specific to this task and not shared with other visual tasks. (3) Other tasks also retain unique variance, which might reflect the task-specific nature of

these activities (e.g., VQ1–VQ3 could involve question-based reasoning, and IU tasks could involve higher-level image understanding.

Similarly, Figures 20 and 21 show Venn diagrams for the early visual and higher visual regions of interest (ROIs), depicting shared and unique variance across these regions. We make the following observations from Figures 20 and 21: (i) Unlike the whole visual cortex, tasks like IC and other task-specific instructions exhibit moderate shared variance in the early visual cortex, while shared variance is significantly higher in higher visual ROIs. This suggests that these tasks depend on similar low-level visual processing mechanisms in early visual areas. (ii) The IU1 instruction ("Describe the most dominant color in the image") shows greater unique variance than IC in the early visual cortex, indicating that low-level color processing is specific to early visual areas, with MLLMs effectively capturing task-specific representations. In contrast, IC exhibits greater unique variance in the higher visual cortex compared to IU1, reflecting the task's reliance on higher-order visual processing.

Overall, these findings demonstrate that shared variance increases from early visual to higher visual areas, reflecting the hierarchical nature of visual processing. Meanwhile, unique variance decreases in higher areas, as tasks rely more on integrated and shared representations. Tasks such as Visual Question Answering (VQ) and Visual Reasoning (VR) retain distinct processing features across both early and higher ROIs, underscoring their unique demands on visual and cognitive processing. These results highlight the capacity of MLLMs to distinguish task-specific representations and align closely with brain visual processing mechanisms.

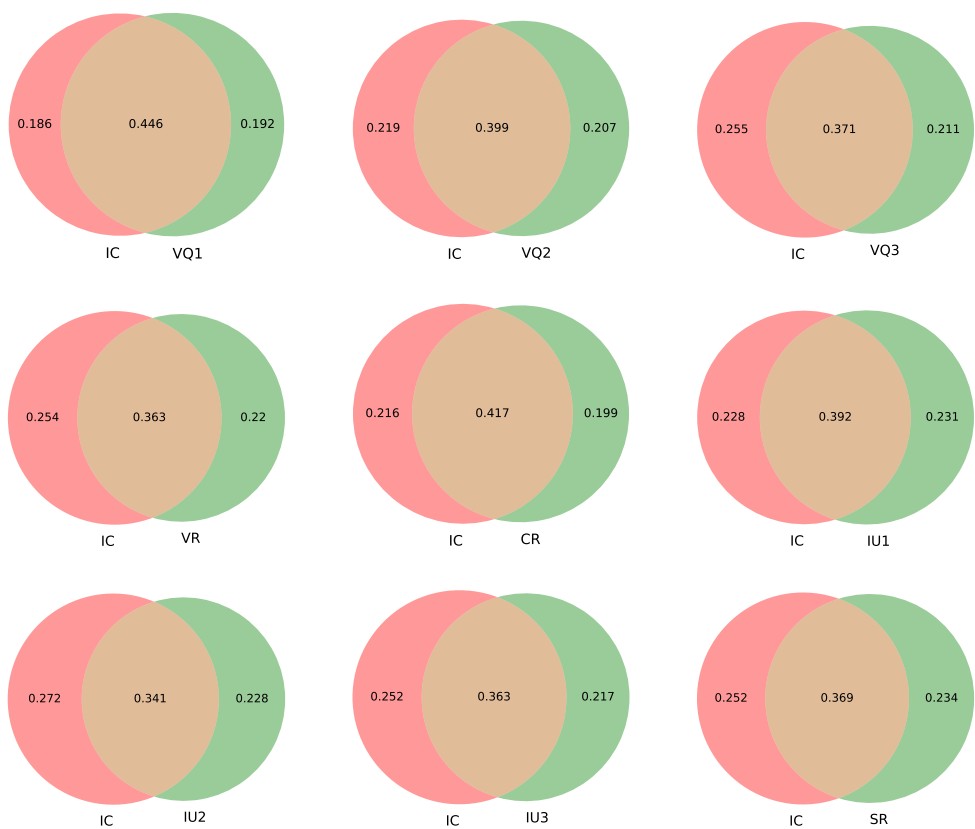

Figure 19: Whole Visual Cortex: Shared and Unique Variance explained between task instructions: Image Captioning (IC) and other task instructions. In each plot, Pink Area (Left Circle - Intersection) represents the unique variance explained by the IC task that is not shared with the corresponding task. Green Area (Right Circle - Intersection) represents the unique variance explained by the corresponding task (e.g., VQ1, CR, etc.) that is not shared with the IC task. Light Brown Intersection (Overlap) represents the shared variance between the IC task and the corresponding task. It indicates the extent to which both tasks explain overlapping neural variance in the visual cortex.

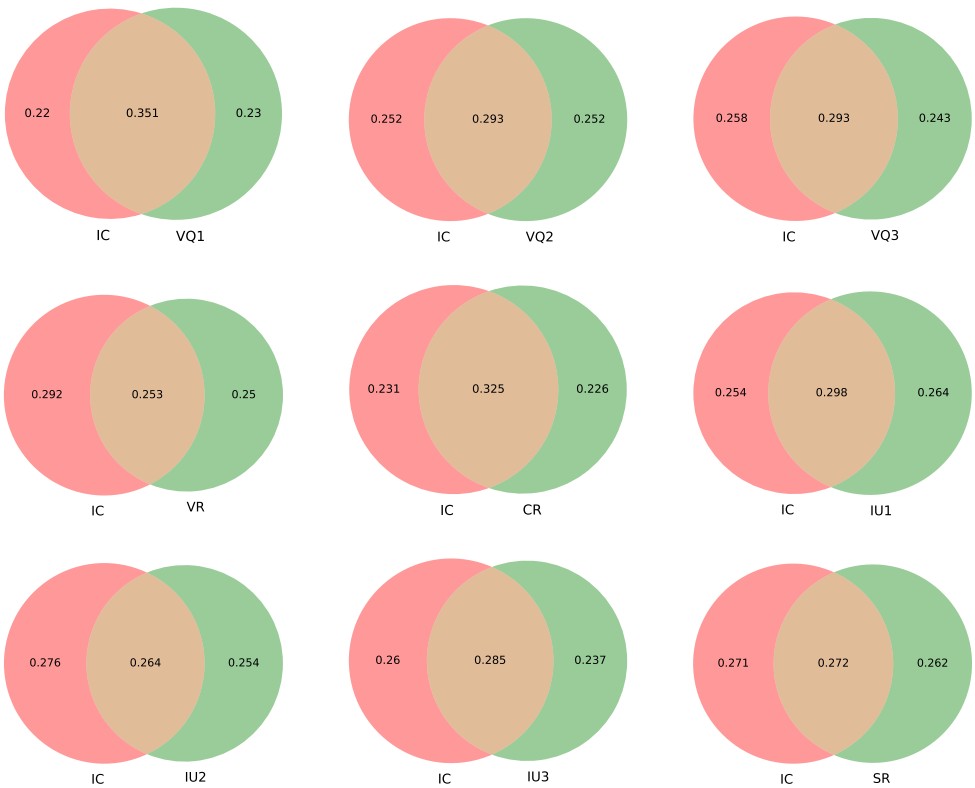

Figure 20: Early Visual Cortex: Shared and Unique Variance explained between task instructions: Image Captioning (IC) and other task instructions. In each plot, Pink Area (Left Circle - Intersection) represents the unique variance explained by the IC task that is not shared with the corresponding task. Green Area (Right Circle - Intersection) represents the unique variance explained by the corresponding task (e.g., VQ1, CR, etc.) that is not shared with the IC task. Light brown Intersection (Overlap) represents the shared variance between the IC task and the corresponding task. It indicates the extent to which both tasks explain overlapping neural variance in the visual cortex.

## M   IMAGE ONLY / INSTRUCTION ONLY INPUT TO THE INSTRUCTION-TUNED MLLM

We performed two additional experiments to investigate the behavior of instruction-tuned MLLMs, as reported in Fig. 22: **Image-Only Input with Empty Prompt:** Here, the input to the instruction-tuned MLLM consists only of images, with an empty string provided as the instruction prompt. From this experiment, we observed that the embeddings generated by the instruction-tuned MLLM using only images and an empty prompt perform similar to the CLIP-image embeddings. Our hypothesis is that providing an empty instruction effectively reduces the instruction-tuned MLLM to behave similarly to a vision-language model like CLIP, resulting in comparable brain encoding performance to the CLIP model. This supports the notion that the absence of an instruction prompt shifts the model's behavior towards a more vision-centric embedding generation. **Instruction-Only Input with Empty Image:** In contrast to image only input, when only task instructions are provided with no image input, the embeddings perform below the randomly initialized baseline. This demonstrates that visual input is crucial for achieving meaningful brain alignment, as MLLMs with visual input, such as "InstructBLIP" (vision+language) and "InstructBLIP No Prompt" (vision-only), significantly outperform the instruction-only baseline.

Fig. 22 illustrates the normalized brain alignment across the whole visual cortex, comparing the InstructBLIP model with and without a prompt, without image, as well as the ViT-H and CLIP-Text models. This figure shows that InstructBLIP with a prompt achieves higher normalized brain alignment compared to the model without a prompt. Furthermore, even when passing a single modality

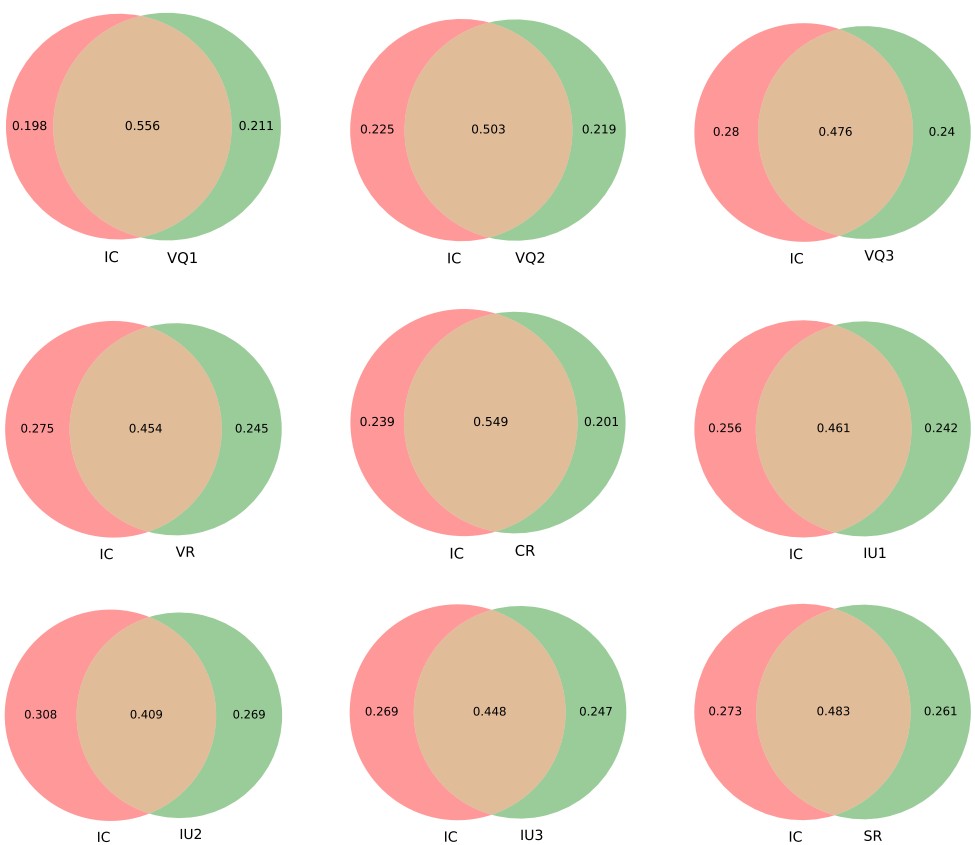

Figure 21: Higher Visual Cortex: Shared and Unique Variance explained between task instructions: Image Captioning (IC) and other task instructions. In each plot, Pink Area (Left Circle - Intersection) represents the unique variance explained by the IC task that is not shared with the corresponding task. Green Area (Right Circle - Intersection) represents the unique variance explained by the corresponding task (e.g., VQ1, CR, etc.) that is not shared with the IC task. Light brown Intersection (Overlap) represents the shared variance between the IC task and the corresponding task. It indicates the extent to which both tasks explain overlapping neural variance in the visual cortex.

(image) to the InstructBLIP model, it retrieves relevant embeddings from the text modality within its aligned embedding space, similar to the behavior of the CLIP model. This is likely due to the model being pretrained with images and their associated task-specific instructions, enabling it to leverage its multimodal alignment. However, when only task instructions are provided without an image input, the embeddings perform below the randomly initialized baseline, emphasizing the critical role of visual input in achieving meaningful alignment. This occurs because the InstructBLIP model is specifically designed to integrate both visual and textual information. Without the image input, the model loses a critical part of the visual context, severely affecting its ability to comprehend and generate accurate responses. In scenarios with both image and instruction inputs, the model can leverage the visual context to better interpret and respond to the instructions. In contrast, the absence of an image deprives InstructBLIP of this advantage, leading to significantly poorer performance compared to conditions where both inputs are provided. Overall, these results highlights the dependency of MLLMs on visual input for robust performance and supports the importance of including both visual and task-specific instructions for achieving high brain alignment.

## N  LIMITATIONS

Apart from the generic limitations of fMRI such as poor temporal resolution (compared to EEG or MEG) and the blood oxygen level dependent (BOLD) signal being an indirect measure of neural activity (Logothetis, 2008), a specific limitation of the current study is that it relies on the NSD

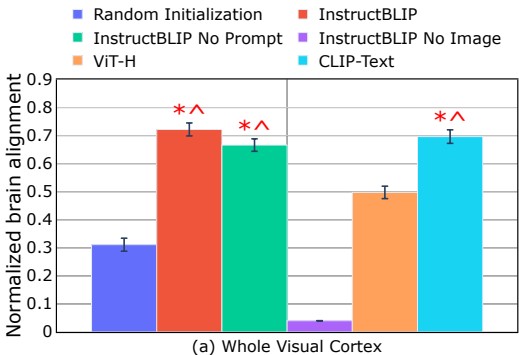

Figure 22: Whole Visual Cortex: Comparison of Instruction-tuned MLLMs performance (Instruct-BLIP with and without prompt, without image).

dataset, where subjects passively viewed images. As a result, the dataset may not fully capture how brain activity aligns with task-specific instructions. Collecting brain recordings while subjects engage in tasks guided by different instructions could provide a more comprehensive evaluation of whether instruction-tuned MLLMs truly reflect visual information processing in response to natural instructions. It is also important to note that while we observed several task-specific instructions leading to improved brain alignment between fMRI recordings and MLLMs, not all instructions were relevant for brain alignment. This indicates that our analysis may not have fully captured the range of task-specific instructions that are jointly processed by the brain. Future work could expand on our approach by incorporating additional task-specific instructions to better characterize the joint processing of information between the brain and instruction-tuned MLLMs.

Since the NSD dataset was collected while participants engaged in watching scenes, our work primarily focuses on the visual areas of the brain, as these are most relevant for the stimuli and tasks studied. Additionally, leveraging the associated ROI mappings of the NSD dataset, we conducted our analysis across the entire visual cortex. Regions outside the visual cortex, such as the language network, prefrontal cortex, or auditory regions, may not exhibit strong alignment with visual features due to their specialization in non-visual tasks, such as language processing, decision-making, or auditory perception. However, it would be an interesting direction to explore the alignment using instruction-tuned multimodal large language models in scenarios where participants watch videos involving auditory, visual, and language information, enabling a comprehensive whole-brain analysis.

