# OpenReview forum: "Correlating instruction-tuning (in multimodal models) with vision-language processing (in the brain)"
_ICLR.cc/2025/Conference — ICLR 2025 Poster_

### Official Review · Reviewer_vZ7u · 2024-10-22

**Soundness:** 3
**Presentation:** 3
**Contribution:** 3
**Rating:** 8
**Confidence:** 3

**Summary:**

The paper studies the correlation between multimodal LLMs (MLLMs) and fMRI from the Natural Scenes Dataset (NSD). Specifically, they feed images and instructions to the MLLMs, cache the embeddings, and fit a linear model to map from MLLM embeddings to fMRI. They find that MLLMs exhibit higher brain alignment than vision-only models and CLIP. They also analyze the correlation of specific instructions and specific MLLM layers with brain regions. Finally, they do a variance partitioning analysis to quantify the overlap between different pairs of instructions.

**Strengths:**

- The central question the paper studies (the alignment of MLLMs and fMRI) is novel, as MLLMs have not been studied in prior work. The paper is also interesting because it presents some insights into the internals of MLLMs, for example, the fact that image captioning overlaps highly with other instructions.
- The presentation of the paper is good. The experiments are well-motivated and interpreted with precise language. The paper is also thorough in its setup and appropriately references prior work.
    - The ablations relating specific instructions or model layers to brain regions (Sec. 6.2) were interesting, even if for some instructions the association was inconclusive.
    - Ablations, such as the baseline “cross-subject prediction accuracy” (L174) and “variance partitioning approach” (L262), are motivated by prior work. The usage of a ridge regression based model (L242) and Pearson Correlation as a metric (L252) for brain alignment is also standard practice.

**Weaknesses:**

- The experiments could include additional ablations for the input to the MLLM.
    - For example, similar to the setup in Sec. 6.1, one could ablate feeding only the image or only the instruction to the MLLM, which reduce to “vision-only” or “LLM-only” baselines. This could help control for model size / other model statistics, i.e., ViT-H might also perform worse because it is a smaller model.

**Questions:**

- How does the CLIP baseline work?
    - L235 states “we input both image and ground truth caption pairs” — are the CLIP image and CLIP text embeddings simply concatenated? Are you using the final pooled output from CLIP, or some intermediate layer? If you are using an intermediate layer, how do you pool across image patches / tokens?
- In L29, what does “effectively capture” mean?
    - Does this mean that the MLLM embedding correlates with the “expected” brain region that is known to do a certain type of processing? This is not obvious from the discussion in L365.
- Below are a few minor comments.
    - L280: typo; “random **initialization** of the 3 MLLMs”
- Below is a comment on the limitations section, although this note is not important for my rating and I leave this to the discretion of the authors.
    - I wish the paper would also briefly discuss the limitations of fMRI itself, and how it is not exactly synonymous with “processing in the brain.” Specifically, fMRI is imprecise, as it “measures a surrogate signal” where the “spatial specificity and temporal response” is constrained [1].
    - Nevertheless, fMRI is the most available / accessible brain signal to study, and it is still interesting to study the relation between machine learning models and fMRI.

References

[1] Logothetis, N. What we can do and what we cannot do with fMRI. *Nature* 453, 869–878 (2008).

---

> ### Author Response · Authors · 2024-11-18
>
> *We thank the reviewer for their strong positive, insightful and valuable comments and suggestions which are crucial for further strengthening our manuscript.*
>
> **Q1. The experiments could include additional ablations for the input to the MLLM.  One could ablate feeding only the image or only the instruction to the MLLM**
>
> Thank you for this question.
> * Based on the reviewer's suggestion, we performed an additional experiment where the input to the instruction-tuned MLLM consists only of images, with an empty string provided as the instruction prompt.
> * From this experiment, we observed that the embeddings generated by the instruction-tuned MLLM using only images and an empty prompt perform similar to the CLIP-image embeddings.
> * Our hypothesis is that providing an empty instruction effectively reduces the instruction-tuned MLLM to behave similarly to a vision-language model like CLIP, resulting in comparable brain encoding performance to the CLIP model. This supports the notion that the absence of an instruction prompt shifts the model’s behavior towards a more vision-centric embedding generation.
> * **Appendix N Figure 23** illustrates the normalized brain alignment across the whole visual cortex, comparing the InstructBLIP model with and without a prompt, as well as the ViT-H and CLIP-Text models. This figure shows that InstructBLIP with a prompt achieves higher normalized brain alignment compared to the model without a prompt. Furthermore, even when passing a single modality (image) to the InstructBLIP model, it retrieves relevant embeddings from the text modality within its aligned embedding space, similar to the behavior of the CLIP model. This is likely due to the model being pretrained with images and their associated task-specific instructions, enabling it to leverage its multimodal alignment.
>
> We have added results of these experiments in **Appendix N** of the revised paper.
>
> * As recommended by **Reviewer 3L7C**, we have thoroughly explored the role of instruction tuning and conducted two additional experiments to assess its impact:
>   - Non-instruction-tuned MLLM (BLIP-2): We passed all task instructions as input.
>   - Language Model (LLaMA-2-7B): We passed only captions as input without task instructions.
> * Kindly review the response provided under **Q1 of Reviewer 3L7C**.
> We have added results of these experiments in **Appendix J** of the revised paper.
>
> **Q2. How does the CLIP baseline work?
> L235 states “we input both image and ground truth caption pairs” — are the CLIP image and CLIP text embeddings simply concatenated? Are you using the final pooled output from CLIP, or some intermediate layer? If you are using an intermediate layer, how do you pool across image patches / tokens?**
>
> Thank you for this question.
> * When we input both image and ground truth text caption pairs to the CLIP model, it generates image embeddings and text embeddings, both of the same dimension.
> * We use CLIP-text embeddings to predict brain activity as our focus is more on instruction-tuned generated embeddings from MLLMs.
> * We consider only CLIP-text embeddings, and the CLIP-image embeddings are not used, and the embeddings are not concatenated.
> * Yes, we use the final pooled output from CLIP’s text encoder for our analysis, not any intermediate layer outputs.
>
> **Q3. In L29, what does “effectively capture” mean?
> Does this mean that the MLLM embedding correlates with the “expected” brain region that is known to do a certain type of processing? This is not obvious from the discussion in L365.**
>
> * No, L29 relates to detailed observations in lines 405-413
>
> **Q4. Below are a few minor comments. L280: typo; “random initialization of the 3 MLLMs”**
>
> Thank you for pointing this out. We corrected this typo in the revised draft.
>
> **Q5. I wish the paper would also briefly discuss the limitations of fMRI itself, and how it is not exactly synonymous with “processing in the brain.”**
>
> Thank you for this thoughtful comment. We agree with the reviewer that fMRI signal does not directly measure neural processing but rather provides a surrogate signal reflecting brain activity.
>
> We have added a sentence as a limitation at Line 530 *[logothetis et al. 2008]*.
>
> *[logothetis et al. 2008], What we can do and what we cannot do with fMRI. Nature 453, 869–878 (2008).*

---

> ### Comment · Reviewer_vZ7u · 2024-11-22
>
> Thank you for taking the time to address my concerns and questions. I think the paper is well-done, and a nice contribution investigating the correlations of MLLMs and fMRI.
>
> **Regarding the additional experiment in **Appendix M Figure 23**, I was wondering if the authors could also add the result of "InstructBLIP No Image" (the language-only baseline)**, in addition to "InstructBLIP" (vision+language) and "InstructBLIP No Prompt" (the vision-only baseline)? I think this result would adequately address concerns that the unimodal models are not directly comparable with the VLMs.
>
> Pending this addition, I am prepared to increase my score.

---

> > ### Author Response · Authors · 2024-11-23
> >
> > Dear Reviewer vZ7u,
> >
> > We appreciate your positive feedback and are confident that it has contributed to enhancing the quality of our paper.
> >
> > As per your suggestion, we conducted an additional experiment: InstructBLIP No Image (language-only baseline), where only task instructions were provided without image input. We now report results for InstructBLIP (vision+language), InstructBLIP No Prompt (vision-only baseline), and InstructBLIP No Image (language-only baseline). Appendix M Figure 23 illustrates the normalized brain alignment across the whole visual cortex, comparing these configurations alongside ViT-H and CLIP-Text models.
> >
> > * This figure shows that InstructBLIP with an image + prompt achieves higher normalized brain alignment compared to the model without a prompt.
> > * Furthermore, even when passing a single modality (image) to the InstructBLIP model, the “InstructBLIP No Prompt” model retrieves relevant text modality signals from its aligned embedding space. This is likely due to the model being pretrained with images and their associated task-specific instructions, enabling it to leverage its multimodal alignment.
> > * In contrast, the language-only baseline (task instructions without images) performs below the randomly initialized baseline, highlighting the critical role of visual input in meaningful alignment.
> >
> > Overall, these results highlight the dependency of MLLMs on visual input for robust performance and supports the importance of including both visual and task-specific instructions for achieving high brain alignment.

---

> > > ### Comment · Reviewer_vZ7u · 2024-11-23
> > >
> > > Thank you for adding the requested baseline; I have reviewed the updated figure and increased my score.
> > >
> > > I would recommend that that authors look further into why the result for InstructBLIP No Image is so much worse than the Random Initialization and add some explanation in Sec. N of the Appendix.
> > >
> > > Regardless, my concerns have been addressed and the authors have satisfactorily ablated the role of multimodality with regards to brain alignment.

---

> > > > ### Author Response · Authors · 2024-11-25
> > > >
> > > > We appreciate the reviewer's positive feedback and are confident that it has enhanced the paper's quality.
> > > >
> > > > In response to your recommendation, we have included additional explanation in Appendix N of the revised draft.

---

### Official Review · Reviewer_6bnp · 2024-10-31

**Soundness:** 3
**Presentation:** 3
**Contribution:** 3
**Rating:** 6
**Confidence:** 4

**Summary:**

The authors test the ability of instruction tuned multimodal LLMs to match human visual cortex responses to static scenes in the NSD. They compare instruction tuned models to one standard multimodal model (CLIP) and a unimodal vision model.

The findings reveal that all multimodal models match visual cortex responses better in both low- and high-level visual regions. There does not seem to be a difference between CLIP and instruction-tuned models in any region. Within the instruction tuned models, there seem to be differences between tasks that best explain different parts of visual cortex, with lower-level tasks (e.g., color) better describing retinotopic areas and higher-level tasks (e.g., image captioning) better explaining higher-level tasks. Variance partitioning reveals that there is shared variance between many pairs of tasks, but some tasks like food labeling and scene recognition are more unique.

It is difficult to draw strong comparisons about these models compared to unimodal vision models though, because of the many differences (architecture, training data, task). As others have pointed out in prior work, advantages of multimodal models in visual cortex significantly decrease when you consider more balanced pairs (e.g., SLIP family models in Wang et al, and Conwell et al) and it seems likely that many of these differences would also diminish in more controlled modeled comparisons. If there are matched vision transformers trained on the same dataset, including them would strengthen the claims of the paper.

The task comparisons are interesting, but it would help for the authors to spell out what is at stake in these investigations. Does better fit of one type of instruction mean tell us what tasks to train neural networks on for improving human-aligned AI, or does it tell us something about the tuning properties of visual cortex. It would help to clarify this, particularly in the paper introduction and discussion.

**Strengths:**

-	Instruction-tuned multimodal models are an interesting way to investigate task tuning, as they have higher performance than prior fine-tuned models like the taskonomy set
-	The comparison of retinotopic versus category-selective tuning is interesting, and may yield novel insights into high-level vision.
-	Variance partitioning allows a more fine-grained look into the contribution of different task tuning

**Weaknesses:**

-	While the instruction model-brain comparisons are very interesting, it is not entirely clear what is at stake. Is the central question spelled out in the intro (lines 83-85) important for better human-alignment of AI, or in order to reveal tuning properites of the human brain? If the latter, the authors should flesh this out, and state some limits of this model comparison approach (see below). The primary goal of the study should be clarified in the introduction.
-	The authors make a distinction between the advantage of multimodal models in high- versus low-level visual cortex. The results look very qualitatively similar in early versus late ROIs in Figures 2 (and also compared to other category selective regions in Figure 10) so if this is a major point, it should be backed up with a statistical analysis, (e.g., non-parametric anova, or permutation-based comparison of the vision-multimodal difference in both regions).
-	Prior work has shown that when architecture and training set are matched (e.g., the SLIP family models) advantages of multimodal models largely go away (e.g., Wang et al 2023, Conwell et al., 2022). It seems likely that the advantage of the multimodal models over unimodal vision models in this paper is similarly due to different/richer training data, rather than multimodality itself.
-	The distinction between the different instructions/tasks could be made clearer. It seems like the tasks vary from low- high-level and many of the papers claims suggest this distinction. It would help to explicitly order the instructions and make this distinction clear early in the paper (e.g., Table 1). Without a priori labels for low- versus high-level tasks, some of the conclusions seems somewhat circular (eg a task that best explains retinotopic cortex is low-level). This ordering could also help make the results in Figure 3 more clear.
-	It is difficult to see trends that are constant across the two different instruction-tuned models in Figure 3 and 11. InstructBLIP seems to show the trends highlighted in the paper, mPLUG-Owl looks quite random. Perhaps the re-ordering of the tasks/color bar suggested above will help. Alternatively, perhaps small differences across the models are being magnified in the winner-take-all plot. Either way, the authors should address these discrepancies.
-	As a small point, the ROI labels in Figure 2 could be more intuitive. I believe “whole brain” refers to the NSDGeneral mask? If so, this should clarify that it is only visual cortex, not whole brain. pRF-Visual and FLOC-PLACES would be more clear as “retinotopic early visual cortex” and “scene-selective” or something similar
-	It is unclear what Figure 5 is adding to the paper. Layerwise comparisons in different transformer networks are somewhat confusing and also seem dependent on the structure of encoders/decoders. The authors should consider moving this to the supplement. If it is important to the main findings, the author should clarify how the layerwise comparisons relate to the main text, and how they should be interpreted in light of any architectural differences across models (eg encoder vs decoder layers)
-	The variance partitioning is a major strength of the paper, but it seems to only investigate shared variance between pairs of tasks. The more important question seems to be how much (if any) unique variance is explained by models tuned on any one task? And what these results can tell us about the brain.

**Questions:**

-	In Figure 2 it seems like CLIP is far outperforming the randomly initialized network. Why is there no asterisks?
-	Is Figure 6 summarizing the results across NSD-General? Why show only for one representative subject rather than the average of all?

---

> ### Author Response · Authors · 2024-11-18
>
> *We thank the reviewer for their positive, insightful and valuable comments and suggestions which are crucial for further strengthening our manuscript.*
>
> **Q1. The central question spelled out in the intro (lines 83-85) important for better human-alignment of AI, or in order to reveal tuning properities of the human brain?**
>
> Thank you for this question.
> * Since the proposed work is first of its kind, our initial focus was to establish the brain alignment of instruction-tuned MLLMs with task-specific instructions. Therefore the central question important for us is to investigate the feasibility of a better human-alignment of AI. We have clarified this now in the introduction as suggested.
>
> **Q2. The authors make a distinction between the advantage of multimodal models in high- versus low-level visual cortex. The results should be backed up with a statistical analysis, (e.g., non-parametric anova)?**
>
> Thank you for pointing this out.
> * We agree that a quantitative comparison is necessary to substantiate the distinction between the performance of multimodal models in early and late visual ROIs.
> * Based on your suggestion, we conducted a non-parametric one-way ANOVA test to analyze the normalized brain alignment differences between early and higher visual ROIs for each instruction-tuned MLLM across four subjects.
>
> **Results for InstructBLIP Model:**
> * The one-way ANOVA test revealed that higher visual ROIs have significantly higher normalized brain alignment than early visual ROIs, with a p-value of 0.008 and an F-statistic of 14.60.
> * Pairwise Comparisons Between Early Visual and Higher Visual ROIs:
>   - We further performed ANOVA tests between early visual ROIs and each specific higher visual ROI. The results indicate that for most higher visual ROIs, the brain alignment is significantly better compared to early visual ROIs:
>   - Retinotopic Early Visual vs. FLOC-PLACES: p-value: 0.006, F-statistic: 41.45
>   - Retinotopic Early Visual vs. FLOC-Bodies: p-value: 0.001, F-statistic: 32.91
>   - Retinotopic Early Visual vs. FLOC-Faces: p-value: 0.006, F-statistic: 41.02
>   - Retinotopic Early Visual vs. FLOC-Words: p-value: 0.14 (not statistically significant), F-statistic: 2.83
> * These results quantitatively confirm that multimodal models such as InstructBLIP achieve significantly better alignment in higher visual ROIs than in early visual ROIs.
> * We performed a similar analysis to include all instruction-tuned MLLMs and conducted a one-way ANOVA test to compare normalized brain alignment between early visual and higher visual ROIs.
>   - The one-way ANOVA test revealed that higher visual ROIs have significantly higher normalized brain alignment than early visual ROIs, with a p-value of 0.009 and an F-statistic of 13.85.
> We have incorporated these quantitative results into the **Appendix E** in the revised draft.

---

> > ### Author Response · Authors · 2024-11-18
> >
> > **Q3. Prior work has shown that when architecture and training set are matched (e.g., the SLIP family models) advantages of multimodal models largely go away (e.g., Wang et al 2023, Conwell et al., 2022).**
> >
> > Thank you for pointing this out and for giving us the opportunity to clarify our findings in the context of existing research.
> >
> > * *Conwell et al. (2022)* performed controlled comparisons between visual models with the same architecture and training data and found no performance increase as a result of contrastive image-language training. One possible reason for this outcome may be related to the evaluation metrics used in their analysis. *Conwell et al. (2022)* utilized distributional similarity measures such as Representational Similarity Analysis (RSA) even for voxel-wise encoding models. While these metrics are valuable for comparing the overall statistical properties of neural and model representations, they may not capture fine-grained functional correspondences between specific model features and neural responses.
> > * On the other hand, correlation-based metrics like Pearson Correlation Coefficient (PCC) and explained variance are designed to assess the direct relationship between model predictions and neural activity on a voxel-by-voxel basis. These metrics are more sensitive to the nuances in the data and can capture subtle alignments that distributional measures might overlook *[Soni et al. 2024]*.
> > * The difference in evaluation metrics may partially explain why *Conwell et al. (2022)* did not observe a performance increase with contrastive image-language training, whereas other studies, including ours, have found that multimodal models show improved brain alignment as we use correlation-based metrics that are sensitive to fine-grained functional correspondences.
> > * Other studies: *Tang et al. (2023)* and *Wang et al. (2023)* show that multimodal training enhances semantic representations aligned with higher-order brain areas. *Oota et al. (2022)* and *Nakagi et al. (2024)* confirm that multimodal models uniquely capture brain activity in the association cortex better than unimodal models. All these studies use correlation as an evaluation metric and explain the variance through brain predictivity.
> > * Further, it is to be noted that the representations extracted for the instructions from all the models (especially InstructBLIP and IDEFICS) demonstrate similar trends in results although they differ in architecture and training data, thereby pointing out the general nature of representational alignment in the visual cortex.
> >
> > *[Soni et al. 2024] Conclusions about Neural Network to Brain Alignment are Profoundly Impacted by the Similarity Measure, Arxiv 2024*
> >
> > *[Tang et al. 2023], Brain encoding models based on multimodal transformers can transfer across language and vision, NeurIPS-2023*
> >
> > *[Nakagi et al. 2024], Unveiling Multi-level and Multi-modal Semantic Representations in the Human Brain using Large Language Models, EMNLP-2024*
> >
> > *[Wang et al. 2023], Incorporating natural language into vision models improves prediction and understanding of higher visual cortex, Nature Machine Intelligence 2023*
> >
> > *[Oota et al. 2022], Visio-Linguistic Brain Encoding, COLING-2022*
> >
> > **Q4. The distinction between the different instructions/tasks could be made clearer. Explicitly ordering the instructions and presenting this distinction clearly early in the paper, such as in Table 1, would improve readability and understanding.**
> >
> > Thanks for this suggestion. We have now updated Table 1 in the revised paper, by ordering the tasks by complexity, from least to most complex.
> >
> > **Q5. Re-ordering of the tasks/color bar helps in trends that are constant across the two different instruction-tuned models?**
> >
> > Thank you for this question.
> > * Following the reordering of tasks, we updated the brainmaps for both the InstructBLIP and mPLUG-Owl models, as shown in **Appendix K, Fig. 19**. However, this reordering did not significantly affect the brainmap visualizations, as the updated plots closely resemble the earlier brainmaps presented in Fig. 3 of the main paper. Overall, the results remain consistent with the inherent performance characteristics of the mPLUG-Owl model.
> > * We observed that both InstructBLIP and IDEFICS show the trends highlighted in the paper. We also observe in Fig 5 that mPLUG-OWL behaves differently compared to InstructBLIP and IDEFICS. (i.e., the middle layers of InstructBLIP and IDEFICS models show greater brain alignment for higher visual regions, whereas the later layers are more aligned with early visual regions. In contrast, the later layers of mPLUG-Owl result in higher brain alignment for both higher and early visual regions.)

---

> > > ### Author Response · Authors · 2024-11-18
> > >
> > > **Q6. The ROI labels in Fig. 2 could be more intuitive.**
> > >
> > > Thank you for this valuable suggestion.
> > > * We have updated the ROI labels in revised draft to improve clarity and alignment with commonly understood terminology:
> > >   - Whole brain -> Whole Visual Cortex.
> > >   - We did not change pRF-Visual to be consistent with the naming convention in the NSD dataset. But we have clarified this in Section 3 of the revised draft (See Line 173).
> > >   - The other region names (Places, Words, Faces) remain unchanged to ensure consistency and ease of interpretation, in alignment with the NSD dataset.
> > >
> > > Hope that is ok?
> > >
> > > **Q7. Layerwise comparisons in different transformer networks are somewhat confusing and also seem dependent on the structure of encoders/decoders.**
> > >
> > > Thank you for this question.
> > > * First, we would like to clarify that all language models employed in MLLMs are decoder-based models.
> > > * Our findings indicate that representations from the middle layers of the InstructBLIP and IDEFICS align more closely with brain-relevant visual semantics, whereas the later layers align more closely in models such as mPLUG-Owl. This observation aligns with prior linguistic brain encoding studies *[Toneva et al. 2019, Caucheteux et al. 2022, Oota et al. 2023]*, which consistently show that middle layers of language models exhibit stronger brain alignment compared to earlier or later layers.
> > > * One hypothesis for this phenomenon is that the initial layers of MLLMs primarily process token embeddings, focusing on basic lexical or syntactic information. The later layers, in contrast, are more influenced by task-specific adjustments and the constraints imposed by the model’s objective function (e.g., language generation). The middle layers, however, strike a balance by providing richer, contextually grounded representations, making them better suited for capturing brain-relevant semantics.
> > > * This insight has significant implications for the AI community. It highlights the need to interpret MLLMs more deeply to understand what types of information are encoded at each layer and how this information is hierarchically processed. Such understanding could pave the way for improved architectures and training paradigms that better align with the brain’s representational structures and processing mechanisms.
> > >
> > > But even now, if the reviewer feels that this is better discussed in the appendix, we are happy to do so.
> > >
> > > *[Toneva et al. 2019], Interpreting and improving natural-language processing (in machines) with natural language-processing (in the brain), NeurIPS-2019*
> > >
> > > *[Caucheteux et al. 2022], Language processing in brains and deep neural networks: computational convergence and its limits, Nature Communications Biology, 2022*
> > >
> > > *[Oota et al. 2023], Joint processing of linguistic properties in brains and language models, NeurIPS-2023*
> > >
> > >
> > > **Q9. In Figure 2 it seems like CLIP is far outperforming the randomly initialized network. Why is there no asterisks?**
> > >
> > > * Yes, both CLIP (and also ViT-H) outperforms randomly initialized network significantly. As mentioned in the figure caption, the asterisk is included only for MLLMs models.
> > >
> > >
> > > **Q10. Is Figure 6 summarizing the results across NSD-General? Why show only for one representative subject rather than the average of all?**
> > >
> > > * Sorry for the typo; this was a copy-paste mistake. It is actually average across all subjects; it is not just for subject-1. We have corrected this in the revised version.

---

> > > > ### Author Response · Authors · 2024-11-18
> > > >
> > > > **Q8. The variance partitioning is a major strength of the paper, but it seems to only investigate shared variance between pairs of tasks.**
> > > >
> > > > Thank you for this question.
> > > > * We want to clarify that our variance partitioning methodology indeed reports both the shared variance between pairs of tasks and the unique variance explained by models tuned on each task.
> > > >
> > > > **Clarification of our current approach:**
> > > >    - Fig. 6 in the main manuscript illustrates the shared variance between pairs of tasks.
> > > >    - Fig. 7 presents both the unique and shared variances between pairs of task-specific instructions for the InstructBLIP model, focusing on a representative subject (Subject 1) and comparing Image Captioning (IC) versus Image Understanding (IU2).
> > > > * Based on the reviewer's suggestion, we have extended our analysis to explicitly report the unique variance explained by models tuned on individual tasks, not just in the context of task pairs.
> > > > * This expanded analysis is now included in **Appendix Figs. 20, 21 and 22**. In these figures, we present the unique variance captured by each task-specific model across all subjects.
> > > > * **Appendix Fig. 20** shows Venn diagrams representing the shared and unique variance between the IC task and nine other task-based instructions (VQ1, VQ2, VR, CR, IU1, IU2, IU3, SR) across the whole visual cortex. We make the following observations from Fig. 20:
> > > >   - (1) The higher shared variance across most task pairs (e.g., IC \& VQ1, IC \& CR), highlighting that tasks involving visual question answering and captioning share significant neural processing mechanisms in the visual cortex.
> > > >   - (2) IC retains a notable amount of unique variance in most comparisons, suggesting that certain neural processes underlying image captioning are specific to this task and not shared with other visual tasks.
> > > >   - (3) Other tasks also retain unique variance, which might reflect the task-specific nature of these activities (e.g., VQ1–VQ3 could involve question-based reasoning, and IU tasks could involve higher-level image understanding.
> > > > * Similarly,  **Appendix Figs. 21 and 22** show Venn diagrams for the early visual and higher visual regions of interest (ROIs), depicting shared and unique variance across these regions. We make the following observations from Figs. 21 and 22:
> > > >   - (i) Unlike the whole visual cortex, tasks like IC and other task-specific instructions exhibit moderate shared variance in the early visual cortex, while shared variance is significantly higher in higher visual ROIs. This suggests that these tasks depend on similar low-level visual processing mechanisms in early visual areas.
> > > >   - (ii) The IU1 instruction ("Describe the most dominant color in the image") shows greater unique variance than IC in the early visual cortex, indicating that low-level color processing is specific to early visual areas, with MLLMs effectively capturing task-specific representations. In contrast, IC exhibits greater unique variance in the higher visual cortex compared to IU1, reflecting the task's reliance on higher-order visual processing.
> > > > * Overall, these findings demonstrate that shared variance increases from early visual to higher visual areas, reflecting the hierarchical nature of visual processing. Meanwhile, unique variance decreases in higher areas, as tasks rely more on integrated and shared representations.
> > > > * Tasks such as Visual Question Answering (VQ) and Visual Reasoning (VR) retain distinct processing features across both early and higher ROIs, underscoring their unique demands on visual and cognitive processing. These results highlight the capacity of MLLMs to distinguish task-specific representations and align closely with brain visual processing mechanisms.

---

> > > > > ### Comment · Reviewer_6bnp · 2024-11-19
> > > > > **Review responses**
> > > > >
> > > > > Thank you for addressing my concerns. The additional analyses and text edits have strengthened the paper, and I have updated my score accordingly.
> > > > >
> > > > > I felt that two of points in my original review are still open, so I mention them again here:
> > > > >
> > > > > 1.	It is difficult to draw strong comparisons about these models compared to unimodal vision models though, because of the many differences (architecture, training data, task). I appreciate the authors explanation of potential causes for discrepancy in Conwell 2022 based on the RSA metric versus encoding, but the crux of the concern with uncontrolled model comparison still holds.
> > > > > 2.	It is difficult to see trends that are constant across the two different instruction-tuned models in Figure 3 and 11. InstructBLIP seems to show the trends highlighted in the paper, mPLUG-Owl looks quite random. Perhaps the re-ordering of the tasks/color bar suggested above will help. Alternatively, perhaps small differences across the models are being magnified in the winner-take-all plot. Either way, the authors should address these discrepancies.

---

> > > > > > ### Author Response · Authors · 2024-11-20
> > > > > >
> > > > > > Dear Reviewer 6bnp,
> > > > > >
> > > > > > Thank you for addressing our rebuttal and for updating your score.
> > > > > >
> > > > > > We appreciate the reviewer’s strong positive feedback and are confident that it has contributed to enhancing the quality of our paper.
> > > > > >
> > > > > > * We acknowledge the two points you raised regarding the reordering of tasks and associated brain maps plot, as well as the discussion on uncontrolled model comparisons. We will address these points promptly during this discussion period.
> > > > > >
> > > > > > **Re-ordering of the tasks/color bar helps in trends that are constant across the two different instruction-tuned models?**
> > > > > >
> > > > > > Thank you for this question.
> > > > > > * For the reordering of tasks, we have updated the brainmaps for both the InstructBLIP and mPLUG-Owl models, as shown in Appendix K, Fig. 19. However, this reordering did not significantly affect the brainmap visualizations, as the updated plots closely resemble the earlier brainmaps presented in Fig. 3 of the main paper. Overall, the results remain consistent with the inherent performance characteristics of the mPLUG-Owl model.
> > > > > > * We observed that both InstructBLIP and IDEFICS show the trends highlighted in the paper. We also observe in Fig 5 that mPLUG-OWL behaves differently compared to InstructBLIP and IDEFICS. (i.e., the middle layers of InstructBLIP and IDEFICS models show greater brain alignment for higher visual regions, whereas the later layers are more aligned with early visual regions. In contrast, the later layers of mPLUG-Owl result in higher brain alignment for both higher and early visual regions.)
> > > > > >
> > > > > > Regards,
> > > > > >
> > > > > > The Authors

---

### Official Review · Reviewer_WsrE · 2024-11-02

**Soundness:** 3
**Presentation:** 3
**Contribution:** 3
**Rating:** 6
**Confidence:** 2

**Summary:**

The paper investigates the alignment of instruction-tuned multimodal Large Language Models (MLLMs) with brain activity, particularly focusing on how these models process visual information when prompted with natural language instructions. The study explores the brain alignment of MLLMs compared to unimodal and non-instruction-tuned multimodal models and assesses the effectiveness of various task-specific instructions on brain alignment. The paper presents a comprehensive analysis of how different instructions from MLLMs capture visual concepts and their correlation with brain activity, using fMRI data from the Natural Scenes Dataset (NSD).

**Strengths:**

- The paper introduces a novel approach to evaluating the brain alignment of instruction-tuned MLLMs, providing valuable insights into how these models process multimodal information in relation to human brain activity.
- The findings have implications for understanding how brain activity corresponds to the processing of multimodal information, which could be valuable for cognitive neuroscience and AI research.

**Weaknesses:**

- The study relies on the NSD dataset, where subjects passively view images, which may not fully capture brain activity aligned with task-specific instructions. Active task engagement during fMRI scans could provide a more comprehensive evaluation.
- How do the authors address ethical considerations regarding the use of fMRI data, especially in relation to participant privacy and data security?

**Questions:**

None

---

> ### Author Response · Authors · 2024-11-18
>
> *We thank the reviewer for their strong positive, insightful and valuable comments and suggestions which are crucial for further strengthening our manuscript.*
>
> **Q1. The study relies on the NSD dataset, where subjects passively view images, which may not fully capture brain activity aligned with task-specific instructions. Active task engagement during fMRI scans could provide a more comprehensive evaluation.**
>
> Thank you for this question.
> * We agree. We had already included this in our limitations paragraph on Page 10 of the original paper. Unfortunately no datasets exist where participants are involved in the corresponding task. All existing datasets involve participants passively watching images. We agree that collecting such data is a great direction for future work.
>
> **Q2. How do the authors address ethical considerations regarding the use of fMRI data, especially in relation to participant privacy and data security?**
>
> Thank you for this question.
> * We agree with the reviewer that it is crucial to understand the ethical landscape surrounding human brain research.
> * We use the publicly available NSD dataset, where the collection and use of neuroimaging data involve significant privacy and ethical considerations, as mentioned in [Allen et al., 2022]. The NSD dataset is publicly available at http://naturalscenesdataset.org.
> * Further, The NSD fMRI dataset we use is a well-known public dataset that has previously been used in many publications in both ML and neuroscience venues *(ML [Andrew et al. 2023, Scotti  et al. 2023, Scotti et al. 2024, Andrew et al. 2023, Sarch et al. 2023: , Neuroscience: [Wang et al. 2022])*.
>
> *[Andrew et al. 2023], Brain Diffusion for Visual Exploration: Cortical Discovery using Large Scale Generative Models, NeurIPS-2023*
>
> *[Andrew et al. 2023], BrainSCUBA: Fine-Grained Natural Language Captions of Visual Cortex Selectivity, ICLR-2024*
>
> *[Scotti et al. 2023], Reconstructing the Mind's Eye: fMRI-to-Image with Contrastive Learning and Diffusion Priors, NeurIPS-2023*
>
> *[Scotti et al. 2023], Shared-Subject Models Enable fMRI-To-Image With 1 Hour of Data, ICLR-2024*
>
> *[Sarch et al. 2023], Brain dissection: fMRI-trained networks reveal spatial selectivity in the processing of natural images, NeurIPS-2023*
>
> *[Wang et al. 2023], Incorporating natural language into vision models improves prediction and understanding of higher visual cortex, Nature Machine Intelligence-2023*

---

> > ### Author Response · Authors · 2024-11-27
> >
> > Dear Reviewer WsrE,
> >
> > We appreciate your feedback and effort you have invested in evaluating our work.
> >
> > We have carefully addressed the questions you raised regarding reliance on NSD dataset and ethical consideration. We kindly request you to verify our response and consider updating your evaluation based on the revisions made.
> >
> > Should you have any further questions or suggestions, we are ready to provide additional information or clarification as needed.
> >
> > Thanks for your help.

---

### Official Review · Reviewer_3L7C · 2024-11-02

**Soundness:** 3
**Presentation:** 3
**Contribution:** 3
**Rating:** 8
**Confidence:** 4

**Summary:**

The paper studies brain-encoding fits of instructions-tuned MLLMs in comparison with traditional methods for designing multimodal embeddings such as via CLIP. To study this question, the paper feeds different types of instructions to the MLLMs in comparison to other baselines like CLIP or vision encoders. The paper finds that instruction-tuned MLLMs have much better fits with visual areas of the brain than vision embedding models and similar performance to CLIP. The authors break down the instructions by type to identify which type of instructions have stronger correlation with visual areas of the brain and do the same with visual concept groups. The authors also include a shared variance experiment that aims to characterize whether shared features were used for each instruction group that were relevant for encoding model performance.

Overall, I liked this paper and would vote for acceptance outside of a few concerns. It answered a relevant question about instruction-tuned MLLMs that haven’t really been explored before.

**Strengths:**

* Novelty with instruction-tuned MLLMs: Overall, I haven’t seen too much work on brain encoding models with instruction-tuned MLLMs. This work is highly timely.
* Well-designed and controlled experiment: The paper uses a controlled and well designed experiment for exploring brain fits.
* Instruction breakdown: I really appreciated Figure 3 and 4. I thought it was pretty interesting to see how different instruction types fit responses in the visual cortex. This was a fairly novel result.
* I also really liked Figure 6 which focused on shared variance. This was an interesting result that showed which instruction types had a higher degree of shared features which resulted in a higher degree of brain similarity.

**Weaknesses:**

* Comparisons: Overall, I think it would be better to include a baseline with a non-instruction-tuned MLLM that has the same architecture. For example, maybe BLIP-2 instead of InstructBLIP? This would have really explored the role of instruction tuning more thoroughly in comparison. BLIP-2 should be available off the shelf and I believe this would be a salient comparison. I would also be curious about how a language model of similar size would do.
* Another concern here is that improvement over ViT-H could be due to an increase in parameters. See questions.
* Nit: Could Figure 3 be sharpened? The text is quite blurry and difficult to read.

Incorporation of some of these comparisons would help raise my score/confidence.

**Questions:**

* The paper focuses on the visual areas of the brain. How would the results look if the entire brain was used? A small discussion would suffice.
* How many parameters are ViT-H and CLIP? Are they comparable?

---

> ### Author Response · Authors · 2024-11-18
>
> *We thank the reviewer for their strong positive, insightful and valuable comments and suggestions which are crucial for further strengthening our manuscript.*
>
> **Q1. It would be better to include a baseline with a non-instruction-tuned MLLM that has the same architecture. For example, maybe BLIP-2 instead of InstructBLIP?  I would also be curious about how a language model of similar size would do.**
>
> Thank you for this insightful question.
> * To thoroughly explore the role of instruction-tuning, as suggested by the reviewer, we conducted two additional experiments:
>   - Non-instruction-tuned MLLM (BLIP-2): We passed all task instructions as input.
>   - Language Model (LLaMA-2-7B): We passed only captions as input without task instructions.
>
> * We have added results of these experiments in **Appendix J** of the revised paper.
>
> * **Firstly**, we present the generated **output tokens from the BLIP-2 model in Appendix Table 6**. This table includes examples of the generated tokens for each task instruction. We already provided predictions from InstructBLIP for the same image in Appendix Table 3. From the examples in Table 6, we observe that the generated output tokens adhere more closely to captioning instructions, regardless of the specific task instructions provided. The outputs often consist of simple responses, such as "Yes," "No," or color names, and lack detailed descriptions. In contrast, instruction-tuned MLLMs excel at providing semantically rich and conceptually grounded descriptions, as shown in Table 3, demonstrating a significant difference in their ability to follow task-specific instructions effectively.
>
> * **Second**, we present the **scatter plot in Appendix Fig. 17**, which compares the performance of **InstructBLIP vs. BLIP-2** for brain predictivity. The plot includes all voxels across the visual cortex with normalized brain alignment scores, where the diagonal represents identical performance for both models.
>   - Image Captioning Task (left): The histogram shows a distribution of voxels deviating from the diagonal towards InstructBLIP, indicating that InstructBLIP performs better. However, the deviation is more pronounced for voxels with normalized brain alignment scores > 0.2.
>   - Visual Relation Task (right): Similarly, the voxel distribution deviates significantly towards InstructBLIP, demonstrating its superior performance. The deviation is notably larger for this task compared to the image captioning task.
>
> * **Third**, in Appendix Fig. 18, we extend the analysis from Fig. 2 of the main paper by computing normalized brain alignment for the following regions:
>   - (a) Whole Visual Cortex,
>   - (b) Early Visual Cortex  (pRF-Visual), and
>   - (c) High-level Visual Cortex (Bodies, Faces, Places, Words)
>
> * The results, now included in the Appendix (Fig. 18), demonstrate that the non-instruction-tuned MLLM (BLIP-2) achieves brain alignment scores (averaged across all task instructions) that are marginally below those of the CLIP-Text model.
> * In contrast, the LLaMA-2-7B model demonstrates performance closer to the ViT-H model. This behavior is attributed to the fact that non-instruction-tuned models generate output tokens that adhere more closely to captioning instructions, regardless of specific task instructions.
>
> **Key Findings:**
> * The performance boost in instruction-tuned MLLMs is primarily due to their ability to generate semantic descriptions and conceptually understand the elements of each scene, as opposed to merely generating task-specific answers, as seen in non-instruction-tuned MLLMs.
> * Representations from the LLaMA-2-7B model, which are based solely on captions, exhibit more semantic information but lack visual understanding details. Consequently, its performance aligns more closely with visual model representations (e.g., ViT).
>
> **Conclusion:**
> * Instruction-tuned MLLMs demonstrate superior visual understanding when provided with task-specific instructions, as evidenced by their higher brain alignment. This reinforces the value of instruction tuning in enhancing multimodal understanding and alignment with human brain representations.

---

> ### Author Response · Authors · 2024-11-18
>
> **Q2. The paper focuses on the visual areas of the brain. How would the results look if the entire brain was used? A small discussion would suffice.**
>
> Thank you for this insightful question.
> * Since the NSD dataset was collected while participants engaged in watching scenes, our work primarily focuses on the visual areas of the brain, as these are most relevant for the stimuli and tasks studied. Additionally, leveraging the associated ROI mappings of the NSD dataset, we conducted our analysis across the entire visual cortex.
> * Regions outside the visual cortex, such as the language network, prefrontal cortex, or auditory regions, may not exhibit strong alignment with visual features due to their specialization in non-visual tasks, such as language processing, decision-making, or auditory perception. However, as suggested by the reviewer, it would be an interesting direction to explore the alignment using instruction-tuned multimodal large language models in scenarios where participants watch videos involving auditory, visual, and language information, enabling a comprehensive whole-brain analysis.
>
> We have added this discussion in **Appendix M.1** of the revised draft and consider it as an exciting direction for future work.
>
> **Q3. How many parameters are ViT-H and CLIP? Are they comparable?**
>
> Thank you for this question.
>
> * The ViT-Huge (ViT-H) is indeed one of the largest Vision Transformer variants, consisting of 632M total parameters.
> * In our work, we utilize the CLIP model with the same ViT-H backbone, which consists of 986M total parameters—632M for the image encoder and 354M for the text encoder.
> * While the vision backbones of the two models are identical and directly comparable, the overall model parameters are not entirely comparable due to the added multimodal capabilities and parameters of CLIP.
>
> **Q4. Another concern here is that improvement over ViT-H could be due to an increase in parameters.**
>
> Thank you for this insightful question.
>
> * While it is true that the CLIP model has a higher total parameter count than ViT-H due to the addition of the text encoder (986M total vs. 632M for ViT-H), the improvement observed in our results is not solely attributable to the increased parameters.
> * Recent brain encoding studies that explore parameter scaling in vision models *[Matsuyama et al., 2022]* and language models *[Antonello et al., 2023]* reveal that while scaling parameters can improve prediction accuracy according to scaling laws, the improvement is often marginal and depends on the sample size.
> * In contrast, the increased brain alignment observed in the CLIP model compared to the ViT-H model is primarily due to its multimodal nature, which leverages joint vision-language representations. This multimodal capability enables the model to encode richer, more context-aware features that are particularly beneficial when aligning with brain activity.
> * We observe that the CLIP model with the ViT-B backbone (151M parameters: 87.85M for the image encoder and 63.43M for the text encoder) outperforms the ViT-H model (632M parameters), achieving a normalized brain alignment score of 0.64 across the whole visual cortex compared to 0.51 for the ViT-H model, despite having significantly fewer parameters. This superior performance is attributed to the multimodal rich representations leveraged by the CLIP model.
> * Therefore, the enhanced brain alignment in the CLIP model is not just due to an increase in parameters but rather mainly because of the additional semantic information obtained from the text modality.
>
> **Q5. Could Fig. 3 be sharpened? The text is quite blurry and difficult to read.**
>
> Thank you for this valuable suggestion.
> * We have updated the color bar in Fig. 3 to make it clearer and easier to read.
>
> *[Matsuyama et al., 2022], Applicability of scaling laws to vision encoding models, Arxiv 2023*
>
> *[Antonello et al., 2023], Scaling laws for language encoding models in fMRI, NeurIPS-2023*

---

> > ### Comment · Reviewer_3L7C · 2024-11-22
> > **Reviewer Response**
> >
> > Thank you for addressing my concerns. I will raise my score since I think this is a valuable contribution but have a few additional comments:
> > 1. Figure 3: I still find the text in figure 3 really hard to read since it's too bright on the page. Could a different color scheme be used where the colors aren't so bright?
> > 2. I generally agree with reviewer 6bnp that some more discussion on the performance of mPLUG-Owl would be very useful. The trends are a bit difficult to see but I think having some additional text would be useful. That being said, I think the other findings are useful.

---

> > > ### Author Response · Authors · 2024-11-25
> > >
> > > We appreciate the reviewer's positive feedback and are confident that it has enhanced the paper's quality.
> > >
> > > * Regarding the reordering of tasks, we have already updated the brainmaps for both the InstructBLIP and mPLUG-Owl models in an earlier revision, as shown in Appendix K, Fig. 19. However, this reordering did not significantly affect the brainmap visualizations, as the updated plots closely resemble the earlier brainmaps presented in Fig. 3 of the main paper. Overall, the results remain consistent with the inherent performance characteristics of the mPLUG-Owl model.
> > > * We acknowledge the reviewer's suggestion to improve the color scheme for better contrast. This valuable feedback will further strengthen the quality of the paper.

---

### Author Response · Authors · 2024-11-18
**Summary of our responses and revision:**

*We are grateful to all reviewers for their strong positive feedback, time and their constructive suggestions, which will further strengthen the impact of our work.*

**Summary of Reviewer Strengths:**

1. Novelty for brain alignment with instruction-tuned MLLMs: Timely and innovative research. Provides insights into how these models process multimodal information in relation to human brain activity (**Reviewer 3L7C, WsrE, 6bnp, vZ7u**)
2. Well-designed and controlled experiment. (**Reviewer 3L7C**)
3. Instruction breakdown:
   - how different instruction types fit responses in the visual cortex, a novel result. (**Reviewer 3L7C**)
   - shared variance, showing which instruction types have higher shared features and brain similarity. (Reviewer 3L7C, 6bnp**)
4. Implications for cognitive neuroscience and AI research:
   - Findings help understand how brain activity corresponds to processing multimodal information. (**Reviewer WsrE**)
5. Good presentation: Well-motivated and precisely interpreted experiments. (**Reviewer vZ7u**)
6. Interesting ablations: Relates specific instructions or model layers to brain regions. (**Reviewer vZ7u**)


**Additional changes to the draft during the rebuttal process**

We have updated the main manuscript and the appendix to address these following comments. The changes made in the manuscript are highlighted in blue color. The major additional changes are listed below.

1. **Two additional experiments on comparison of Instruction-tuned MLLMs, Non-Instruction-tuned MLLMs and text-based LLMs. (Reviewer-3L7C)**:
   - Non-instruction-tuned MLLM (BLIP-2): We passed all task instructions as input.
   - Language Model (LLaMA-2-7B): We passed only captions as input without task instructions.
We have added results of these experiments in **Appendix J** of the revised paper.

2. **Quantitative comparison between early and high-level visual ROIs in instruction-tuned MLLMs (Reviewer-6bnp)**:
   -  We conducted a non-parametric one-way ANOVA test to analyze the normalized brain alignment differences between early and higher visual ROIs for each instruction-tuned MLLM across four subjects.
   - The one-way ANOVA test revealed that higher visual ROIs have significantly higher normalized brain alignment than early visual ROIs, with a p-value of 0.009 and an F-statistic of 13.85.

3. **Re-ordering of the task-specific instruction and brainmaps (Reviewer-6bnp)**:
   - We have now updated Table 1 in the revised paper, by ordering the tasks by complexity, from least to most complex.
   - Following the reordering of tasks, we updated the brainmaps for both the InstructBLIP and mPLUG-Owl models, as shown in **Appendix K, Fig. 19**.
   - However, this reordering did not significantly affect the brainmap visualizations, as the updated plots closely resemble the earlier brainmaps presented in Fig. 3 of the main paper

4. **Whole Visual Cortex and ROI Analysis: Shared and Unique variance across task-specific instructions (Reviewer-6bnp)**:
   - We have extended our analysis to explicitly report the unique variance explained by models tuned on individual tasks, not just in the context of task pairs.
   - This expanded analysis is now included in **Appendix Figs. 20, 21 and 22**. In these figures, we present the unique variance captured by each task-specific model across all subjects.

5. **Extended Discussion (Reviewer-WsrE, 3L7C)**:
   - Since the NSD dataset was collected while participants engaged in watching scenes, our work primarily focuses on the visual areas of the brain, as these are most relevant for the stimuli and tasks studied. Additionally, leveraging the associated ROI mappings of the NSD dataset, we conducted our analysis across the entire visual cortex.
   - Studying alignment of instruction-tuned MLLMs on whole visual cortex

6. **Image only / Instruction only input to the instruction-tuned MLLM (Reviewer-vZ7u)**:
   - We performed an additional experiment where the input to the instruction-tuned MLLM consists only of images, with an empty string provided as the instruction prompt.
   - **Appendix N Figure 23** illustrates the normalized brain alignment across the whole visual cortex, comparing the InstructBLIP model with and without a prompt, as well as the ViT-H and CLIP-Text models. This figure shows that InstructBLIP with a prompt achieves higher normalized brain alignment compared to the model without a prompt.

---

### Meta-Review · Area_Chair_i7eA · 2024-12-16

**Metareview:**

This work investigates the alignment of instruction-tuned multimodal large language models (MLLMs) with brain activity, specifically focusing on their fit with visual cortex responses compared to traditional baselines like CLIP and unimodal vision models. Using fMRI data, the paper demonstrates that instruction-tuned MLLMs exhibit superior alignment with visual brain areas, comparable to CLIP, while outperforming vision-only models. The analysis also uncovers insights into the effect of instruction specificity on brain alignment and shared variance across tasks.

The work received positive feedback from four reviewers, who commended its novelty in exploring brain alignment with instruction-tuned MLLMs. During the discussion phase, the authors actively addressed reviewers’ concerns by incorporating additional experiments, including comparisons among instruction-tuned MLLMs, non-instruction-tuned MLLMs, and text-based LLMs. They also provided a quantitative analysis of early versus high-level visual ROIs in instruction-tuned MLLMs, reordered task-specific instructions with corresponding brain maps, and conducted comprehensive analyses of the whole visual cortex and ROI-specific shared and unique variances across task-specific instructions. Further, they explored inputs with image-only and instruction-only modalities for the instruction-tuned MLLM.

The Area Chair (AC) agreed with the reviewers that this task is novel and that the proposed experimental settings and findings could significantly benefit future research. The authors have committed to releasing their code and were advised to improve the color scheme for better contrast, ensuring clear and consistent conclusions in experiments.

**Additional Comments On Reviewer Discussion:**

Authors have summarized the points raised by reviewers in response, and each point was carefully considered in the decision.
The paper makes a significant contribution by exploring a novel intersection of AI and neuroscience.
The authors made substantial and timely revisions, conducting new experiments and improving the presentation. Most concerns were satisfactorily addressed, with the remaining points being minor or reasonable for future work.
The added experiments and insights from the rebuttal further strengthened the study’s claims and utility for future research.

---

### Decision · Program_Chairs · 2025-01-22

Accept (Poster)